**Brief Communication**

# Predictive analyses of regulatory sequences with EUGENe

Adam Klie [1,2], David Laub[1,2], James V. Talwar[1,2], Hayden Stites[3], Tobias Jores [4], Joe J. Solvason[1,2,5], Emma K. Farley[1,2,5] & Hannah Carter [1,2] ✉

Deep learning has become a popular tool to study cis-regulatory function. Yet efforts to design software for deep-learning analyses in regulatory genomics that are findable, accessible, interoperable and reusable (FAIR) have fallen short of fully meeting these criteria. Here we present elucidating the utility of genomic elements with neural nets (EUGENe), a FAIR toolkit for the analysis of genomic sequences with deep learning. EUGENe consists of a set of modules and subpackages for executing the key functionality of a genomics deep learning workflow: (1) extracting, transforming and loading sequence data from many common file formats; (2) instantiating, initializing and training diverse model architectures; and (3) evaluating and interpreting model behavior. We designed EUGENe as a simple, flexible and extensible interface for streamlining and customizing end-to-end deep-learning sequence analyses, and illustrate these principles through application of the toolkit to three predictive modeling tasks. We hope that EUGENe represents a springboard towards a collaborative ecosystem for deep-learning applications in genomics research.

Cracking the cis-regulatory code that governs gene expression remains a fundamental challenge in genomics research. Efforts to annotate the genome with functional genomics data[1] have powered machine learning methods that aim to learn biologically relevant sequence features by directly predicting these readouts. Deep learning has become especially popular in this space, and has been successfully applied to tasks such as DNA and RNA protein binding motif detection[2-6], chromatin state prediction[7-18], transcriptional activity prediction[10,19-22] and 3D contact prediction[23-26]. Complementary models have recently been developed to predict data from massively parallel reporter assays that directly test the gene regulatory potential of selected sequences[27-29]. Most encouragingly, many of these multilayered models go beyond state of the art predictive performance to generate expressive representations of the underlying sequence that can be interpreted to better understand the cis-regulatory code[16,27,30].

Despite these advances, executing a deep-learning workflow in genomics remains a considerable challenge. Although model training has been substantially simplified by dedicated deep-learning libraries

such as PyTorch[31] and Tensorflow[32], nuances specific to genomics data create an especially high learning curve for performing analyses in this space. On top of this, the heterogeneity in implementations of most code associated with publications greatly hinders extensibility and reproducibility. These conditions often make the development of genomics deep-learning workflows painfully slow even for experienced deep-learning researchers, and potentially inaccessible to many others.

Accordingly, the genomics deep-learning community has assembled software packages[33-37] that aim to address one or more of these challenges. However, each toolkit on its own does not offer both end-to-end functionality and simplicity, and there remains a general lack of interoperability between packages. For instance, Kipoi[36] increases the accessibility of trained models and published architectures, but does not provide a comprehensive framework for an end-to-end deep-learning workflow. Selene[34] implements a library based in PyTorch for applying the full deep-learning workflow to new or existing models, but offers a limited programmatic interface, requires the use of complex configuration files, and has limited functionality for model

[1]Department of Medicine, University of California San Diego, La Jolla, CA, USA. [2]Bioinformatics and Systems Biology Program, University of California San Diego, La Jolla, CA, USA. [3]Daniel Hand High School, Madison, CT, USA. [4]Department of Genome Sciences, University of Washington, Seattle, WA, USA. [5]Department of Molecular Biology, University of California San Diego, La Jolla, CA, USA. ✉e-mail: hkcarter@ucsd.edu

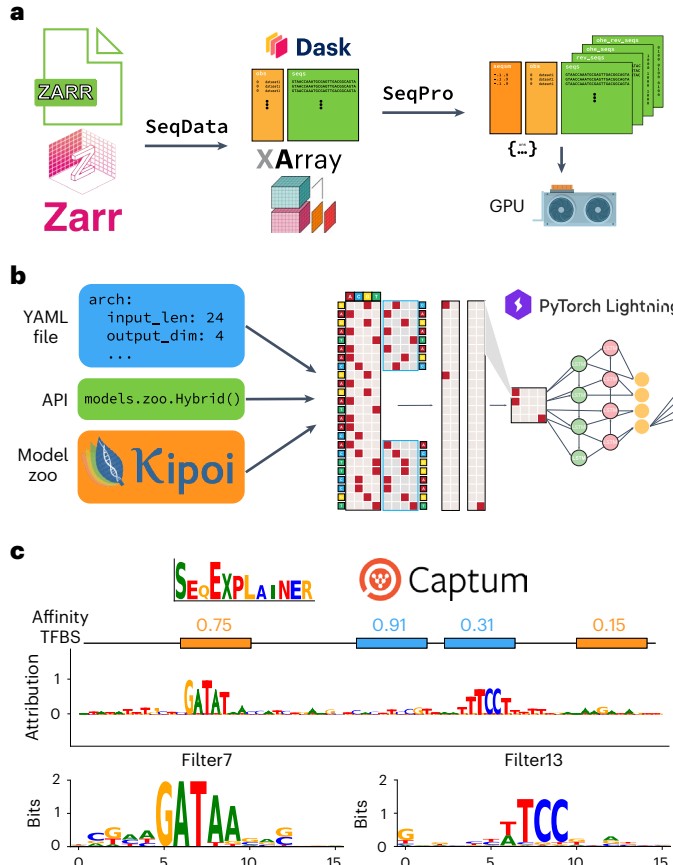

**Fig. 1 | EUGENe workflow for predictive analyses of regulatory sequences.**
**a**–**c**, The EUGENe workflow can be broken up into three primary stages:
data extraction, transformation and loading (ETL) (**a**); model instantiation,
initialization and training (IIT) (**b**); and model evaluation and interpretation
(EI) (**c**). The ETL stage (**a**) begins with using the SeqData subpackage to create
Dask-enhanced XArray datasets backed by Zarr stores. Data transformation
is handled by the SeqPro subpackage, after which data can be loaded into
graphical processing units (GPUs). In the subsequent IIT stage (**b**), model
architectures (such as the example shown in the schematic) are instantiated from
configuration files (in YAML format), from the EUGENe application programming
interface (API), or from Kipoi. EUGENe then uses PyTorch Lightning for training
these architectures. The subpackage SeqExplainer (which is backed by the
Captum package) is used for model interpretation in the EI stage (**c**). Common
visualizations produced by SeqExplainer include the logos depicted for an
example input sequence (top) or for convolutional filters (bottom).

interpretation. Janggu[35], one of the more comprehensive packages,
provides extensive functionality for data loading and for training
models, but offers limited support for PyTorch and limited functional-
ity for model interpretation.

There is generally a need for an end-to-end toolkit in this space that
follows findable, accessible, interoperable and reusable (FAIR) data and
software principles[38], and that is inherently designed to be simple and
extensible. To address this need, we have developed elucidating the
utility of genomic elements with neural nets (EUGENe), a FAIR toolkit
for the analysis of sequence-based datasets.

A standard EUGENe workflow consists of three main stages out-
lined in Fig. 1: extracting, transforming and loading (ETL) data from
common file formats (Fig. 1a); instantiating, initializing and training
(IIT) neural network architectures (Fig. 1b); and evaluating and inter-
preting (EI) learned model behavior on held-out data (Fig. 1c). The
major goal of EUGENe is to streamline the end-to-end execution of
these three stages to promote the effective design, implementation,

validation and interpretation of deep-learning solutions in regulatory
genomics. We have listed several common deep learning for regulatory
genomics tasks that can be implemented in an end-to-end fashion with
EUGENe (Supplementary Table 1). We next describe three in detail,
highlighting the core aspects of the workflow on different data types
and training tasks. A more detailed description of the workflow is
provided in the Methods.

We first used EUGENe to analyze published data from an assay of
plant promoters[29] (Fig. 2a). Jores et al. selected promoter sequences
from −165 to +5-bp relative to the annotated transcription start site
for protein-coding and microRNA genes of *Arabidopsis thaliana*, *Zea
mays* (maize) and *Sorghum bicolor*[29]. A total of 79,838 170-bp promoters
were used to transiently transform two separate plant systems, tobacco
leaves and maize protoplasts. Regulatory activity was quantified using
a variant of the self-transcribing active regulatory region sequencing
(STARR-seq) assay[39] in each system. The resulting data provides two
activity scores that can serve as single task regression targets for train-
ing EUGENe models.

We implemented both the custom BiConv1D layer[40] and convolu-
tional neural network (CNN) architecture (Jores21CNN) described by
Jores and colleagues[29], and then trained separate Jores21CNN architec-
tures for predicting tobacco leaf (leaf models) and maize protoplast
(protoplast models) activity scores. We benchmarked these models
against built-in CNN and Hybrid architectures with matched hyperpa-
rameters, as well as a DeepSTARR architecture[27] (Supplementary Data 1).
As described in the work by Jores et al. (see Methods), we initialized
78 filters of the first convolutional layer of all models with position
weight matrices (PWMs) of plant transcription factors ($n = 72$) and core
promoter elements ($n = 6$)[29]. In both systems, performance metrics
for the most predictive models were comparable with those reported
by Jores and co-workers (Fig. 2b and Supplementary Fig. 1a). We also
trained models on activity scores from both leaves and protoplasts
(combined models) and noted a marked drop in performance (Sup-
plementary Fig. 1b), underscoring differences in the way the leaf and
maize systems interact with the same set of promoters[29].

We next applied several of EUGENe's interpretation functions to
the trained models to determine the sequence features each used to
predict plant promoter activity. First, we used a filter visualization
approach[11] to generate position frequency matrix (PFM) representa-
tions for each of the first convolutional layer's filters and used the
TomTom[41] tool to annotate them. We queried the PFMs against the 78
motifs used to initialize the convolutional layers, both to determine
whether the initialized filters retained their motifs and to see whether
randomly initialized filters learned them de novo. For the leaf and
protoplast models, many of the learned filters were annotated to the
TATA box binding motif and other core promoter elements (Fig. 2c,d).
Only ten learned filters from the combined model were assigned a
significant annotation (adjusted *P*-value < 0.05) by TomTom (Fig. 2d
and Supplementary Fig. 1c), consistent with the observed performance
drop in this system (Supplementary Fig. 1a). We next applied the Deep-
LIFT method[42] to determine the individual nucleotide contributions
for each test set sequence prediction. For many of the sequences with
the highest observed activity scores, the TATA box motifs were often
the lone salient feature identified (Fig. 2e and Supplementary Fig. 1d).
In fact, when only a TATA box motif was inserted into every possi-
ble position in each of the 310 selected promoters, we observed an
147% average increase in predicted activity across insertion positions
and sequence contexts for the leaf model (Fig. 2f and Supplementary
Fig. 1e). Finally, we performed ten rounds of in silico evolution on the
same set of 310 promoters as described in Jores et al. Almost all of the
starting promoters showed a notable increase in predicted activity
after just three mutations (Fig. 2g and Supplementary Fig. 1f). These
results showcase a representative example of the way EUGENe's inter-
pretation suite can be used to identify the key features that a model
uses to make predictions.

To illustrate EUGENe's versatility for different inputs and prediction tasks, we next applied it to analyze RNA binding protein (RBP) specificity data previously introduced by Ray et al.[43] and analyzed through deep learning by Alipanahi and colleagues[2]. In the latter work, they trained 244 CNN models (DeepBind models) that each predicted the binding patterns of a single RBP on a set of 241,357 RNA probes (Extended Data Fig. 1a). The full probe set was designed to capture all possible RNA 9-mers at least 16 times and was split into two balanced subsets (sets A and B) for training and validation, respectively (see Methods)[43]. Each RBP was incubated with a molecular excess of probes from each subset (in separate experiments) and subsequently recovered by affinity purification. The RNAs associated with each RBP were then quantified by microarray and subsequent bioinformatic analysis[44]. This yielded a vector of continuous binding intensity values for each RBP across the probe set that can be used for prediction.

To prepare for training, we first implemented a flexible DeepBind architecture in EUGENe (see Methods) and then trained 244 single task models by using a nearly identical training procedure to Alipanahi et al.[2] (Supplementary Data 2). Along with these single task models, we also randomly initialized and trained a multitask model (Supplementary Data 2) to predict 233 RBP specificities (that is, a 233-dimensional vector) in a single forward pass, excluding 11 RBPs due to a high proportion of missing values across probes in the training set. We also loaded 89 existing Kipoi[36] models trained on a subset of human RBPs in the dataset.

Performance on Set B for all deep-learning models was on par with Set B's correlation to Set A (Extended Data Fig. 1b and Supplementary Fig. 2a) and both single task and multitask models trained with EUGENe showed comparable performance to Kipoi and DeepBind models (Extended Data Fig. 1b,c and Supplementary Fig. 2a,b). The reason for the poor observed performance of certain Kipoi models is not immediately clear, but could relate to differences in sequence or target preprocessing before evaluation. Although the ability to load these pretrained models from Kipoi is very useful for benchmarking, implementing and retraining models is often necessary for fair performance comparisons. EUGENe supports both loading and retraining models, allowing users to more quickly design and execute quality benchmarking experiments.

We next applied EUGENe's interpretation suite to our trained models, first using the filter visualization approach outlined by Alipanahi et al.[2] to generate PFMs for convolutional filters. We again used TomTom to identify filters annotated with canonical RBP motifs[43] in both the best-performing single-task models and the multitask model (Extended Data Fig. 1d and Supplementary Fig. 2c), and found that the number of multitask filters annotated to an RBP was correlated with predictive performance for that RBP (Extended Data Fig. 1d, bottom). We also

calculated attributions for all Set B sequences using the InputXGradient method[42] and observed that canonical motifs were learned by both single- and multitask models (Extended Data Fig. 1e and Supplementary Fig. 2d). Finally, we used EUGENe's sequence evolution functionality to evolve ten random sequences using the single task HNRNPA1L2 model and visualized the attributions for these sequences before and after five rounds of evolution (Extended Data Fig. 1f). Several of the mutations that most increased the predicted score were those that generated canonical binding motifs for the protein. We repeated this for two other RBPs (Pcbp2 and NCU02404) and observed that each model prioritizes mutations that create canonical binding motifs specific to the RBP they were trained on (Supplementary Fig. 2e). These results show that EUGENe simplifies the extraction of salient features from models trained within the same workflow.

As our final use case, we applied EUGENe to the classification of JunD binding as described by Kopp and colleagues[35]. This task uses ChIP-seq data from ENCODE[1] to generate input sequences and binarized classification labels for each sequence (Extended Data Fig. 2a). We used EUGENe to first build a deep-learning-ready dataset for this prediction task (see Methods) and then implemented the CNN architecture described by Kopp et al. (Kopp21CNN). We benchmarked classification performance against built-in fully connected networks (FCNs), CNNs and Hybrid models with matched hyperparameters (Supplementary Data 3). All built-in models were configured to incorporate information from both the forward and reverse strand (double-stranded or 'ds' models).

We trained models using the same procedure described by Kopp et al. (see Methods)[35]. Due to the unbalanced nature of the dataset, we focused on evaluating models with the area under the precision recall curve (AUPRC). For our Kopp21CNNs, we were able to achieve comparable performances on held-out chromosome 3 sequences to those reported by Kopp et al. for one-hot encoded sequences (Extended Data Fig. 2b,c). The dsFCN—the only model without any convolutional layers—immediately overfit the data after a single training epoch and was not predictive of binding (Extended Data Fig. 2c). The dsCNN models, however, achieved higher mean AUPRCs than the dsHybrid models, and much higher AUPRCs than the Kopp21CNN architectures.

We next applied EUGENe's interpretation tools to ask whether our best models were learning sequence features relevant to JunD binding to make predictions. We first generated attributions for the forward and reverse complement strands of all test set sequences using the GradientSHAP[45] method, and visualized the most highly predicted sequences as sequence logos (Extended Data Fig. 2d and Supplementary Fig. 3a). We observed that the most important nucleotides often highlighted consensus or near-consensus JunD motifs, and that these motifs were often attributed similarly on both the forward and reverse

**Fig. 2 | STARR-seq plant promoter activity prediction. a**, jores21 use case schematic. We trained EUGENe models to predict the regulatory activity of 79,838 plant promoters quantified by plant STARR-seq in tobacco and maize. GFP, green fluorescent protein. pA, poly-adenylation site **b**, Performance comparison of four convolution-based architectures on predicting promoter activity in tobacco leaves (left) and maize protoplasts (right). The box plots show distributions of $R^2$ values on held-out test data for each architecture across $n = 5$ independent experiments (random initializations). The boxes show medians along with low and high quartiles. Whiskers extend to the furthest datapoint within 1.5-times the interquartile range. More extreme points are marked as outliers. A two-sided Mann–Whitney U test was used to determine $P$-values, which were adjusted using the Benjamini–Hochberg method (*, adjusted $P$-value < 0.05; ns, not significant). Test statistics and adjusted $P$-values for the leaf models (left) were: CNN–Hybrid ($u = 15$, adjusted $P$-value = 0.10), CNN–DeepSTARR ($u = 24$, adjusted $P$-value = 0.17), CNN–Jores21CNN ($u = 12$, adjusted $P$-value = 1.0), Hybrid–DeepSTARR ($u = 17$, adjusted $P$-value = 1.0), Hybrid–Jores21CNN ($u = 22$, adjusted $P$-value = 1.0), DeepSTARR–Jores21CNN ($u = 14$, adjusted $P$-value = 0.84). Test statistics and adjusted $P$-values for the protoplast models (right) were: CNN–Hybrid ($u = 15$, adjusted $P$-value = 0.03),

CNN–DeepSTARR ($u = 24$, adjusted $P$-value = 0.01), CNN–Jores21CNN ($u = 12$, adjusted $P$-value = 0.01), Hybrid–DeepSTARR ($u = 17$, adjusted $P$-value = 0.01), Hybrid–Jores21CNN ($u = 22$, adjusted $P$-value = 0.01), DeepSTARR–Jores21CNN ($u = 14$, adjusted $P$-value = 0.01). **c**, A hand-selected set of convolutional filters visualized as PWM logos that had significant annotations (adjusted $P$-value < 0.05) to known core promoter elements and transcription factor binding clusters in plants. **d**, Histogram showing the number of learned filters assigned to core promoter elements and transcription factor binding clusters by TomTom with bolded annotations corresponding to the logos in **c**. **e**, Sequence logo visualizations of feature importance scores calculated using the DeepLIFT algorithm on the highest predicted test set sequence in the Hybrid leaf (top) and Jores21CNN protoplast (bottom) models. **f**, Model scores for $n = 310$ sequences implanted with a 16 bp sequence containing a consensus TATA box motif, a shuffled version of the same sequence, an all-zeros sequence and a random sequence (all 16 bp in length). Mean model scores with 95% confidence intervals are shown. **g**, Model scores for the same set of $n = 310$ promoters at different rounds of evolution compared against baseline predictions (evolution round 0). The best Hybrid leaf model was used to generate panels **c**, **d**, **f** and **g** (protoplast model results are shown in Supplementary Fig. 1).

strands (Extended Data Fig. 2d and Supplementary Fig. 3a); however, there were instances in which salient motifs were highlighted on one strand but not the other (Extended Data Fig. 2d), indicating the utility of incorporating information from both strands for prediction. We next generated PFM representations for all ten filters of each convolutional model (excluding dsFCNs) and annotated them using TomTom against the HOCOMOCO FULL v.11 database[46] (Extended Data Fig. 2e and Supplementary Fig. 3b). Among the top hits, we found several filters annotated with motifs such as JunD and CTCF (Extended Data Fig. 2e and Supplementary Fig. 3b). Finally, we performed an in silico experiment with the best overall model, in which we slid a consensus

JunD motif across each position of a set of ten randomly generated sequences and predicted binding (Extended Data Fig. 2f). We observed that the simple inclusion of the consensus binding site led to a considerable jump in predicted output with some position specificity. These results once again showcase that EUGENe's interpretation methods can help explain model predictions, in this case for DNA protein binding from a genome-wide assay.

There are numerous opportunities for future development of EUGENe, but we see a few as high priority. EUGENe is primarily designed to work on nucleotide sequence input (DNA and RNA), but currently does not have dedicated functions for handling protein sequence or

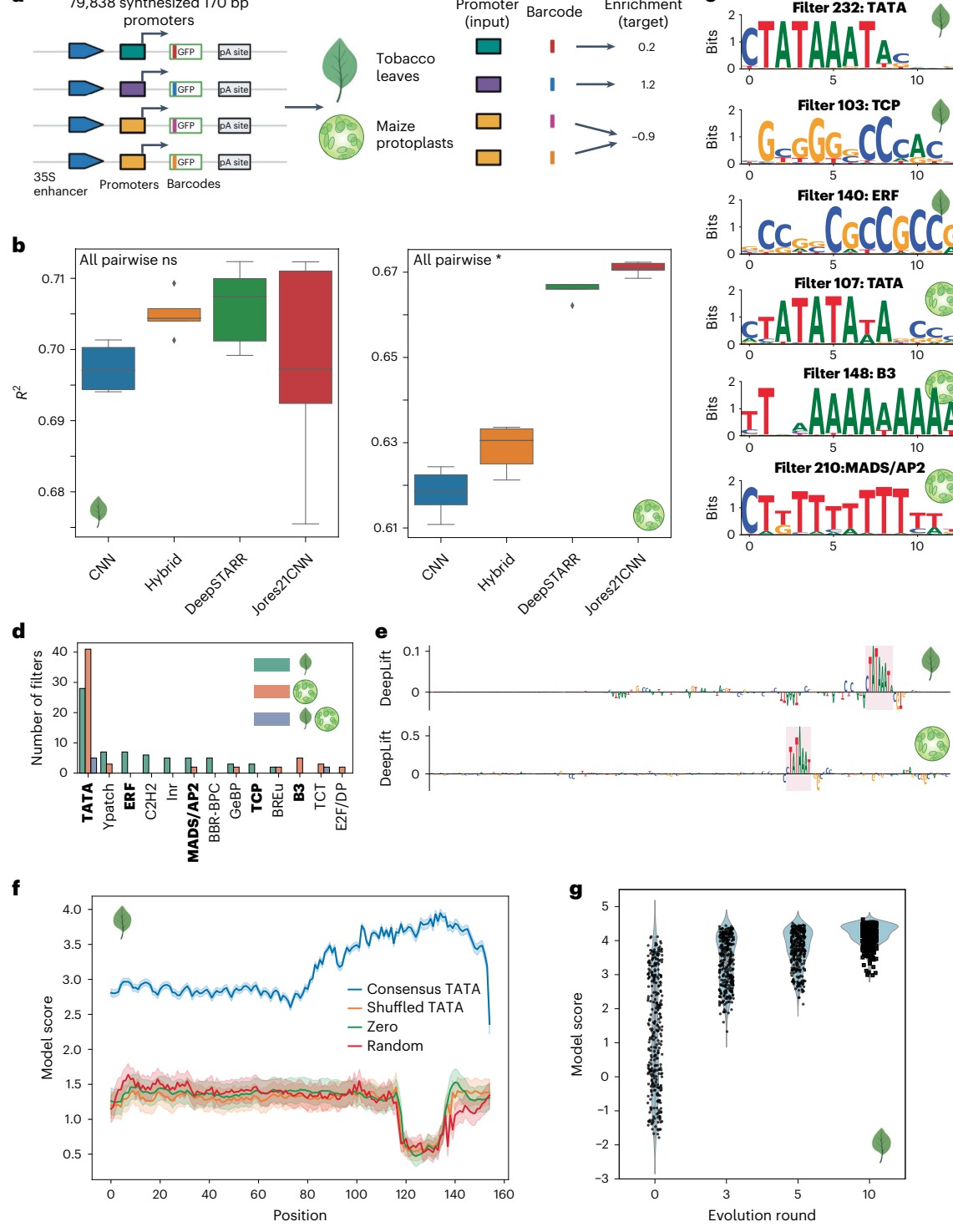

multimodal inputs. Furthermore, as assays move from bulk to single-cell resolution, it will be important to develop functionality for handling single-cell data that allows users to easily ask questions about cell-type-specific regulatory syntax. Finally, we plan on expanding EUGENe's dataset and model library to encompass a larger portion of those available in the field.

The heterogeneity in data types and methods that exist in deep learning for regulatory genomics and the rapid pace with which the field advances makes maintaining FAIR software in this space a major challenge. One of the tasks in Supplementary Table 1, for instance, involves a recently developed and highly specific data formatting and preprocessing pipeline[47]. The use of bespoke methods for data preprocessing, as well as for model interpretation, is quite common in the field, and is often necessary to train accurate models that avoid common machine learning pitfalls[48]. For example, some workflows may require complex implementations of train and test set splitting to protect against information leakage[49]. We see substantial value in continuing to extend EUGENe into spaces such as these, and have designed the toolkit to allow for easy integration of this type of functionality. To continue to make bespoke methods and workflows accessible, we intend to encourage community development of EUGENe through tutorials, workshops and a dedicated user group.

As large consortia (such as ENCODE Phase 4 and Impact of Genomic Variation on Function) and individual groups continue to generate functional genomics data at both the bulk and single-cell level, the need for a standardized deep-learning analysis ecosystem to investigate complex relationships in this data becomes even more pressing. We believe that EUGENe represents a positive step in the direction of such an ecosystem and will empower computational scientists to rapidly expand their knowledge, develop and share methods and models, and answer important questions about the genome and how it encodes function.

## Methods

### The EUGENe workflow

**Data extraction, transformation and loading with SeqData.** The EUGENe workflow begins with extracting data from on-disk formats. Although standardized file formats exist in regulatory genomics, their complexity can make creating model-ready datasets non-trivial. To address this in EUGENe, we created the standalone subpackage, SeqData[50], which flexibly and efficiently reads data from a variety of file formats, including CSV/TSV (tabular), FASTA, BED, BAM and BigWig (Extended Data Fig. 3a, top). The versatility of SeqData enables the generation of many custom datasets from combinations of these file types, including several commonly used in regulatory genomics. These include: (1) datasets derived from combinations of tabular and FASTA files that are suitable for single- and multitask regression and classification (for example, DeepSTARR[27]); (2) datasets from genomic coordinates defined in BED files suitable for multitask binary classification (such as DeepSEA[7] or Sei[15]); and (3) datasets from multiple BigWigs and BED files suitable for binned or base-pair resolution regression (for example, Basenji[10] and BPNet[30], respectively). EUGENe also supplies a growing collection of hand-curated datasets available via the SeqDatasets subpackage[51] (Supplementary Data 4) that can be downloaded and subsequently loaded into a workflow via a single function call (Extended Data Fig. 3a, bottom).

By default, SeqData reads files from disk as XArray datasets[52] backed by Zarr stores[53] (Fig. 1a). We chose to use XArray and Zarr as they are scalable, capable of handling high-dimensional data, and have been previously used in a variety of bioinformatics domains[54–56]. Furthermore, Zarr stores can be loaded out-of-core thanks to functionality offered by XArray and Dask[57], allowing for processing and training of large-scale datasets (Supplementary Fig. 4a). As is standard in deep learning, training in EUGENe is always performed by loading data into GPU memory in batches (when a GPU is available), but is slowed by using the out-of-core functionality on the CPU (Supplementary Fig. 4b). Thus, the decision on whether to first load the dataset into CPU memory before training should balance the available resources and dataset size. Certain datasets, such as those used to train Enformer[21] or Basenji[10], will probably require this out-of-core functionality; however, we have found that many useful and large datasets can entirely fit into memory on machines with less than 32 GB of RAM (Supplementary Data 5).

Once created, an array of functions can be called directly on these XArray datasets to perform common preprocessing steps. EUGENe includes a baseline set of functions for train and test set splitting (for example, by chromosome, fraction or homology[58]) and target normalization (for example, binning, Z-score, clamping and so on) (Extended Data Fig. 3b, left). Sequence preprocessing is handled by the SeqPro subpackage[59], which includes Numba-accelerated[60] padding and one-hot encoding of DNA and RNA sequences (Extended Data Fig. 3b, right), as well as jittering and k-mer frequency-preserving shuffling[61]. EUGENe also fully supports data visualization through the Matplotlib[62] and Seaborn[63] libraries (Extended Data Fig. 3c) and conversion of XArray datasets to formats ingestible by deep-learning frameworks in a highly flexible manner (Extended Data Fig. 3d). Finally, XArray datasets can easily be converted to more familiar Python data structures (NumPy arrays, Pandas DataFrames and so on) and back to allow the user to access the functionality of these libraries.

**Model training with PyTorch and PyTorch Lightning.** Designing and training neural networks for regulatory genomics requires a comprehensive library of architecture building blocks. EUGENe builds on the PyTorch library of neural network layers by adding several useful layers such as inception and residual layers. Furthermore, EUGENe provides flexible functions for instantiating common 'blocks' and 'towers' that are composed of heterogeneous sets of layers arranged in a predefined or adaptable order. For instance, a convolutional block (Conv1DBlock in EUGENe) often comprises convolutional, normalization, activation and dropout layers in different orderings depending on the model and task (Extended Data Fig. 3e, top). On top of this, EUGENe's library supports customizable fully connected (FCN), convolutional (CNN), recurrent (RNN) and Hybrid (a combination of the three, shown in Fig. 1b) architectures that can be instantiated from single function calls or configuration files (Extended Data Fig. 3e, bottom, and Supplementary Data 6, basic architectures). We have also constructed several published architectures that represent specific configurations of these basic architectures, and made them accessible to users through single function calls (Supplementary Data 6). Users looking to use their own custom architectures can also do so, as EUGENe only requires that an architecture be defined by its layers ('init' function) and how inputs are propagated through those layers (forward function; Supplementary Fig. 5a). In summary, model architectures can be instantiated from the application programming interface (API), built from scratch using our library, or imported from external repositories or packages. We provide a detailed tutorial on instantiating architectures via these different mechanisms in EUGENe's tutorials repository[64].

Once instantiated, architectures can be initialized with parameters sampled from standard initialization distributions (Extended Data Fig. 3f, top), or in the special case of convolutional filters, initialized with known motifs[8,16] (Extended Data Fig. 3f, bottom). EUGENe can then be used to fit initialized architectures to datasets (with the option to perform hyperparameter optimization through the RayTune package[65]), and to assess performance and generalizability on held-out test data (Extended Data Fig. 3g). For training, EUGENe uses the PyTorch Lightning framework[66] and programmatic objects called LightningModules. Each EUGENe LightningModule delineates the architecture types it can train and standardizes boilerplate tasks for those architectures (for example, optimizer configuration, metric logging and so on). For instance, the primary LightningModule in EUGENe, termed SequenceModule (Extended Data Fig. 3h), anticipates

training an architecture that takes in a single tensor (typically one-hot encoded DNA sequences) and delivers a single tensor output. We have also implemented a ProfileModule for BPNet-style[30] training, in which models produce multiple tensor outputs (or 'heads'), accept optional control inputs and use multiple loss functions[67]. Using LightningModules in this manner requires only minor code modifications to allow for the reuse of the same architectures in different training schemes and tasks (Supplementary Fig. 5b) and for the fine-tuning of pretrained models (Supplementary Fig. 5c). We plan on continuing to develop the library of LightningModules for different training schemes, including adversarial learning[68], generative modeling[69], language modeling[70] and more.

**Model interpretation with SeqExplainer.** Interpreting models is of critical importance in regulatory genomics[71–73], but is often made challenging by the complexity of neural networks and methods for their interpretation. To address this in EUGENe, we created a standalone subpackage called SeqExplainer that makes various post-hoc interpretation strategies accessible to most PyTorch models trained on one-hot encoded genomic sequences[74]. SeqExplainer currently provides functionality for filter interpretation, attribution analysis, in silico experimentation and sequence generation. Each strategy is briefly detailed below.

The interpretation of learned convolutional filters, commonly employed for model architectures that begin with a convolutional layer, involves using the set of sequences that activate a given filter (maximally activating subsequences) to generate a PFM (Extended Data Fig. 3i). The PFM can then be converted to a PWM, visualized as a sequence logo, and annotated with tools such as TomTom[41], using databases of known motifs such as JASPAR[75] or HOCOMOCO[46]. Filter interpretation in this manner does have limitations. TomTom can be inaccurate when annotating motifs from learned filters[18,76] and this analysis does not specify the importance of each filter for model predictions[76]. Despite these limitations, filter interpretation can be useful for hypothesis generation and for further exploration of how architecture affects learned representations[76–78].

Attribution analysis involves using the trained model to score every nucleotide of the input on how it influences the downstream prediction for that sequence (Extended Data Fig. 3j). In SeqExplainer and EUGENe, we currently implement several common attribution approaches. These include standard in silico saturation mutagenesis, InputXGradient[42], DeepLIFT[42] and GradientSHAP[45], with the last three using functionality from the Captum package[79]. Attributions can also be used to validate that the model has learned representations that resemble motifs. Unlike the filter interpretability approach described above, attributions are directly linked to model predictions, and can naturally be extended to model the effects of single-nucleotide polymorphisms; however, attributions represent a 'local', often noisy[80,81] interpretation of a single sequence, and can require clustering into 'global' attributions for cleaner interpretation. In SeqExplainer we offer wrappers for running the popular TF-MoDISco algorithm[82] to accomplish this.

Attribution analysis, although very useful, stops short of quantifying the effect of whole motifs on model predictions. To get at the quantitative effects of such patterns, EUGENe offers a wide range of functionality for conducting in silico experiments with motifs of interest[27,30], also known as global importance analyses (GIAs)[5]. As the space of possible GIAs is essentially infinite, and the type of GIA used is often dependent on the data, model and biological question being asked, we provide the building blocks for GIAs in SeqExplainer, including functionality for generating background sequences and introducing perturbations (for example, mutations, motif embedding, motif occlusion and so on) to those sequences (Extended Data Fig. 3k). EUGENe currently offers high-level functions for streamlining positional importance analysis (Extended Data Fig. 3l) and distance-dependent motif

cooperativity analysis, and we anticipate adding many more common GIAs to EUGENe in the future.

The last class of interpretability methods currently offered in EUGENe uses trained models to guide sequence evolution. We implement the simplest form of this approach that iteratively evolves a sequence by greedily inserting the mutation with the largest predicted impact at each iteration. Starting with an initial sequence (for example random, shuffled and so on), this strategy can be used to evolve synthetic functional sequences[69] (Extended Data Fig. 3m). This style of analysis is a promising direction for further research, and can also be used for validating that the model has learned representations that resemble motifs.

### Analysis of plant promoter data
**Data acquisition and preprocessing.** Plant promoter assay data were obtained from the GitHub repository associated with the work by Jores and co-workers[29]. These included two identical libraries for a set of 79,838 plant promoters synthesized with an upstream viral 35 S enhancer and downstream barcode tagged GFP reporter gene (Fig. 2a). The libraries were designed to include 10–20 constructs with distinct barcodes for each promoter. These libraries were used to transiently transform both tobacco leaves and maize protoplasts and promoter activities were assayed using plant STARR-seq[39]. Per-barcode activity was calculated as the ratio of RNA barcode frequency to DNA barcode frequency and the median of these ratios was then used to aggregate across barcodes assigned to the same promoter. These aggregated scores were then normalized by the median value for a control construct and were log transformed to calculate a per-promoter 'enrichment' score. We downloaded these enrichment scores[83] for both libraries as separate datasets which we could use as training targets. We used the identical 90/10 training and test split used in Jores et al. (the dataset could be downloaded with set labels). The training set was further split into 90/10 train and validation sets. All sequences were one-hot encoded using a channel for each letter of the DNA alphabet ('ACGT').

**Model initialization and training.** We implemented the Jores21CNN architecture by translating the Keras code in the associated GitHub repository into PyTorch and integrating it into our library. We benchmarked this architecture against built-in CNN, Hybrid and DeepSTARR architectures in EUGENe with the hyperparameters described in Supplementary Data 1. In each convolutional layer, the Jores21CNN first applies a set of filters to the input as is standard for convolutional models, but also applies the reverse complements of the filters (as opposed to the reverse complement of the sequences) to each input in an effort to capture information from both strands[40]. As this still only requires a single strand as input into the models, we opted to benchmark against only single-stranded versions of built-in CNN and Hybrid models. Following instantiation, we initialized 78 filters in the first convolutional layer of each model using PWMs derived from core promoter elements and transcription factor binding clusters downloaded from the GitHub repository[84] associated with the publication. All of the other parameters were initialized by sampling from the Kaiming normal distribution[85]. We trained models for a maximum of 25 epochs with a batch size of 128 and used the Adam optimizer with an initial learning rate of 0.001. We also included a learning rate scheduler that modified the learning rate during training with a patience of two epochs. We used mean squared error as our objective function and stopped training early if the validation set error did not decrease after five epochs.

**Model evaluation and interpretation.** Models were primarily evaluated using the percentage of variance explained ($R^2$) on predictions for the test set. We repeated the above training procedure across five independent random initializations and evaluated $R^2$ scores across these trials. For PWM visualization, we used the approach described

by Minnoye and colleagues[11]. Briefly, for each filter in the first convolutional layer, we calculated activations for all subsequences (of the same length as the filter) within the test set sequences. We then took the top-100 subsequences corresponding to the top-100 activations (maximally activating subsequences) and generated a PFM. For visualizing filters as sequence logos, we converted PFMs to PWMs using a uniform background nucleotide frequency. We calculated attributions for all test set sequences using the DeepLIFT method[42]. To perform the feature implantation approach, we downloaded the 16 bp PFM containing the consensus TATA box motif from the Jores et al. GitHub repository and one-hot encoded it by taking the highest probability nucleotide at each position. We also downloaded the set of 310 promoters[86] used by Jores et al. for in silico evolution. We then implanted the TATA box containing sequence at every possible position of each of the 310 promoter sequences and used selected high-performing models (one each from leaf, protoplast and combined) to make predictions. We compared this to predicted scores generated with the same feature implantation approach using a shuffled version of the 16 bp sequence containing the TATA box motif, a random 16 bp one-hot encoded sequence, and a 16 bp all zeros input. We performed the in silico evolution experiments on the same set of 310 promoter sequences[29]. In each round, we first used in silico saturation mutagenesis to identify the mutation that increased the model score by the largest positive value (delta score). We then introduced this mutation into the sequence and repeated this for ten iterations.

### Analysis of RNA binding data

**Data acquisition and preprocessing.** As described in detail by Alipanahi and colleagues[2], set of 241,357 31–41 nt long RNA probes were split into two experimental sets (sets A and B), with each designed to include all possible 9-mers at least eight times, all possible 8-mers at least 33 times and all possible 7-mers 155 times (Extended Data Fig. 1a). These probes were assayed against 244 RBPs using a protein binding microarray[44], and intensities were normalized as described by Ray and colleagues[43]. We downloaded the normalized RNA probe binding intensity matrix from the supplementary information of ref. 43, and separated the Set A and B sequences into two distinct groups. To remove outliers, we set all values of probe intensities to be capped at the 99.95 percentile for each prediction task (RBP). We then Z-scored the clamped values to zero mean and unit standard deviation for each RBP. All normalizations were performed using Set A statistics (that is, Set B values were Z-scored using means and standard deviations from Set A). For multitask prediction, we removed the 11 RBPs with ≥0.1% missing values across all probes in Set A, and further removed all probes in Set A that had any missing values for any of the remaining 233 RBPs. This left 120,326 and 110,645 probes for training single- and multitask models, respectively, and 121,031 in Set B for testing. Set A was then further split 80/20 into a training and validation set. All sequences were one-hot encoded using a channel for each of the RNA alphabet ('ACGU') for input into models.

**Model initialization and training.** We implemented the DeepBind architecture described in the supplementary information of Alipanahi et al.[2] and added it as a EUGENe model library. DeepBind architectures were initially designed to take either the forward strand or both strands (ds) as input; however, Alipanahi et al. trained their RBP models with just the single-strand input due to the single-stranded nature of RNA, so we also used a single stranded implementation for our DeepBind models. We initialized both the single task models and the multitask model with parameters sampled from the Kaiming normal distribution[85] and trained all models for a maximum of 25 and 100 epochs, respectively, using the Adam optimizer[87] and an initial learning rate of 0.005. We also included a learning rate scheduler that modified the learning rate during training with a patience of 2 epochs. The batch size for training was fixed to 64 and 1,024 for single- and multitask models,

respectively, and the mean-squared error was used as the objective function for all models, with training halting if the validation set error did not decrease after five epochs. For multitask models, we used the average mean-squared error across all tasks. Hyperparameters selected for the architectures of each model are provided in Supplementary Data 2. Finally, we downloaded a set of 89 pretrained human RBP models[88] from Kipoi and wrapped functions from the Kipoi package to make predictions using these models.

**Model evaluation.** We evaluated models using the Z-score, AUC and E-score metrics reported by Alipanahi and co-workers[2]. To calculate these metrics, we first computed a binary $n \times m$ matrix $A$, where the $n$ rows represent all possible 7-mers from the RNA alphabet (AAAAAAA, AAAAAAC, AAAAAAG and so on), and the $m$ columns represent the 121,031 probes assayed from Set B. Each entry $a_{ij}$ in the matrix is 1 if the $i$th k-mer is found in the $j$th probe and 0 otherwise. Consider first working with a single RBP, in which we have normalized binding intensity values for each of the 121,031 probes ($m$-dimensional vector $\mathbf{x}$). We compared the $i$th row (representing a k-mer) of the matrix $A$ (an $m$-dimensional vector) to the vector $\mathbf{x}$ of observed intensities and computed the Z-scores, AUCs and E-scores for that k-mer as described in ref. 2 and ref. 43. We repeated this for all k-mers (across rows of $A$) to generate an $n$-dimensional vector for each metric meant to capture the importance of each k-mer for binding that RBP. For Z-scores, 0 indicates an average level of binding when that k-mer is present in the probe sequence, with more positive scores indicating higher levels of binding than average when that k-mer is present. For AUC and E-scores (a modified AUC), the value is bound between 0 and 1, with values closer to 1 indicating more binding when that k-mer is present. We repeated this process for all models that predict probe intensities by substituting the predicted intensities from a given model for the vector $\mathbf{x}$ of observed intensities. We generated a set of $n$-dimensional vectors for each model-metric pair (that is, for a single-task model, we have a vector each for the Z-scores, E-scores and AUCs), and then took each of these vectors and calculated Pearson and Spearman correlations with the vector $\mathbf{x}$ from the observed Set B intensities. This results in a pair of correlation values, one Pearson and one Spearman, describing the performance of a given model on a specific RBP (these are single points in the box plots shown in Extended Data Fig. 1b and Supplementary Fig. 2a). Repeating this process for all RBPs generates a distribution of correlations for a given model.

We can use the same procedure on Set A observed intensities to generate a distribution of correlations analogous to a biological replicate. These are the 'Set A' and 'Observed intensities' columns of Extended Data Fig. 1b and Supplementary Fig. 2a. We generated the distribution labeled 'Set A' in this way with our own implementation of these metrics and downloaded the 'Observed intensities' distribution from performance tables included in the supplement of Alipanahi and colleagues. Finally, we calculated Pearson and Spearman correlation coefficients for the observed and predicted intensities on Set B for all models. Note that this is not possible to do for Set A as the probes are different for this set, hence the omission of the 'Set A' and 'Observed intensities' columns in the last box plots of Extended Data Fig. 1b and Supplementary Fig. 2a.

**Model interpretation.** For filter visualization, we used the approach described in Alipanahi and co-workers. Briefly, for a given filter, we calculated the activation scores for all possible subsequences (of the same length as the filter) from Set B probes and identified the maximum value. We then used only the subsequences with an activation at least three-quarters of this maximum to generate a PFM for that filter. We repeated this process for all 16 filters in each of the top-10 single task models and for all 1,024 filters of the multitask model. The top-10 single task models were chosen on the basis of ranking of Pearson correlations between observed and predicted intensity values. We then input all

multitask PFMs to TomTom for annotation against the Ray2013 Homo sapiens database and filtered for hits with a Bonferroni multiple-test-corrected *P*-value ≤ 0.05. We calculated attributions for all Set B probes using the InputXGradient method[42]. For multitask models, attributions can be calculated on a per task basis to determine how each nucleotide of the input sequence influenced that particular task. We again only did this for a subset of RBPs, using the Pearson correlation of predicted and observed intensities to choose the top-10 single task models and the top-10 predicted tasks for the multitask model. We use the same in silico evolution method for this use case as we did for the plant promoters. Using trained models for selected RBPs, we first performed five rounds of evolution on ten randomly generated sequences of 41 nt in length (ACGU sampled uniformly). We then calculated feature attributes for the initial random sequences and the evolved sequences using the InputXGradient method and compared them.

### Analysis of JunD binding data

**Data acquisition and preprocessing.** We followed the same procedure to acquire and preprocess the data for training models on the prediction of JunD binding as reported in a work by Kopp and colleagues[35]. We started by downloading JunD peaks from human embryonic stem cells (H1-hesc) called with the hg38 reference genome from encode-project.org (ENCFF446WOD, conservative IDR thresholded peaks, narrowPeak format). We next defined regions of interest (ROIs) by extending the union of all JunD peaks by 10 kb in each direction. We removed blacklisted regions for hg38 (ref. 89) using bedtools[90] and trimmed the ends of resulting regions to be divisible into 200 bp bins. For training and testing, we binned ROI's into 200 bp sequences and labeled any of those that overlapped a JunD binding peak with a positive label and all non-overlapping bins with a negative label. As input to models, we first extended each genomic bin by 150 bp on each side (so that the model sees 500 bp in total for each input when predicting on a 200 bp site) and then one-hot encoded using a channel for each of the DNA alphabet (ACGT). In total, we used 1,013,080 200 bp bins for generating training, validation and test sets. We split the sequences by chromosome so that validation sequences were from chr2 and test sequences from chr3 (the rest were used for training).

**Model initialization and training.** For the JunD binding task, we first implemented the Kopp21CNN architecture described in a work by Kopp et al.[35] by following the Keras code in the associated GitHub repository along with their description of the layers in the supplementary information of ref. 35. We then trained five random initializations of dsFCNs, dsCNNs, dsHybrids and Kopp21CNNs, each with parameters sampled from the Kaiming normal distribution. All of the models used both the forward and reverse strands as input through the same architecture (ds). Following the work by Kopp and co-workers, we trained all models for a maximum of 30 epochs with the AMSGrad optimizer[91] and an initial learning rate of 0.001. The batch size for training was fixed to 64 for all models and binary cross-entropy was used as the objective function, halting training if the validation set error did not decrease after five epochs. Hyperparameters selected for the architectures of each model are provided in Supplementary Data 3.

**Model evaluation and interpretation.** Models were primarily evaluated using the AUPRC as the dataset was heavily imbalanced. We again performed model interpretation using attributions, filter visualizations and in silico experimentation methods from EUGENe. We calculated attributions for the forward and reverse strands of all test set sequences using the GradientSHAP method[45]. To visualize filters, we applied the approach from ref. 11 and generated PFMs. We fed these PFMs to the TomTom webserver and queried the HOCOMOCO v.11 FULL database[46]. We subset filters down to those with a multiple-test-corrected *P*-value ≤ 0.05 and manually inspected the top hits. These PWMs were visualized as logos using a uniform background of nucleotide frequencies. We performed the in silico implantation experiment using the JunD PFM downloaded from JASPAR[92]. We calculated model scores by generating ten randomly generated sequences (uniformly sampled) and implanting the consensus one-hot encoded JunD motif at every possible position. We compared this to predicted scores from applying the same approach to a random one-hot encoded sequence, an all zeros input and a dinucleotide shuffled JunD motif, all of the same length as the consensus JunD motif.

### Data visualization software

For most exploratory data analysis and performance evaluations, we used a combination of the Seaborn and Matplotlib plotting libraries in Python. For sequence logo visualizations of filters and attributions, we used modified functions from the viz_sequence package[93], and the logomaker package[94].

### Statistical methods

Mann–Whitney U tests[95] were used to compare performance distributions between architecture types and *P*-values were corrected with the Benjamini–Hochberg method[96]. TomTom reports significance of alignments of query motifs to a database using the methods described in ref. 41. We used the *q*-value reported by the webserver tool[97] and considered hits to be those alignments with a *q*-value ≤ 0.05 as significant. Figures for in silico implantation of motifs included 95% confidence intervals.

### Reporting summary

Further information on research design is available in the Nature Portfolio Reporting Summary linked to this article.

## Data availability

All of the datasets used in this study are publicly available. Raw and processed data for the plant promoter STARR-seq were obtained from ref. 83. Normalized RNA probe binding intensities were obtained from ref. 98. JunD peaks from human embryonic stem cells (H1-hesc) called with the hg38 reference genome were obtained from encodeproject.org (ENCFF446WOD, conservative IDR thresholded peaks, narrowPeak format). Blacklisted regions for hg38 were obtained from ref. 89. TomTom queries were performed against the Ray2013 Homo sapiens and the HOCOMOCO v.11 FULL motif collections for the RBP binding and JunD binding use cases, respectively. The JunD PFM was obtained from ref. 92 for the in silico implantation experiment; 89 RBP models were obtained from the Kipoi mode repository at ref. 88. We have also deposited the EUGENe specific dataset files and trained models used in the analyses presented here on Zenodo[99]. These represent the processed data files and SeqData objects that can be used along with the accompanying code to generate the figures for all the use cases. Source Data are provided with this paper.

## Code availability

EUGENe is freely available under the MIT license at https://github.com/ML4GLand/EUGENe. The version of the codebase used for the analyses presented here is available on Zenodo[100]. Documentation for the tool is available at https://eugene-tools.readthedocs.io/en/latest/index.html. Jupyter notebooks and Python scripts used to perform the analyses presented here are available on GitHub at https://github.com/ML4GLand/EUGENe_paper (under the Creative Commons Zero v.1.0 Universal license) and deposited on Zenodo[101].

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

## Acknowledgements

This work was supported by the National Institutes of Health (grant no. 1U01HG012059); infrastructure was funded by the National Institutes of Health (grant no. 2P41GM103504-11); T.J. is supported by the German Research Foundation (DFG; fellowship no. 441540116). E.K.F and J.J.S were supported by the National Institutes of Health (grant no. DP2HG010013). H.C. is supported by the Canadian Institute for Advanced Research (award no. FL-000655). We would like to thank the community of genomics researchers who made their code open source so that we could utilize it for EUGENe functions.

## Author contributions

A.K., J.J.S., E.K.F. and H.C. designed the study. A.K. designed the toolkit. A.K., D.L, J.T. and H.S. implemented the code. A.K. and D.L. performed the use case analyses. J.J.S., D.L., J.T. and H.S. performed software testing. H.C. supervised the work. All authors read and corrected the final manuscript.

## Competing interests

The authors declare no competing interests.

## Additional information

**Correspondence and requests for materials** should be addressed to Hannah Carter.

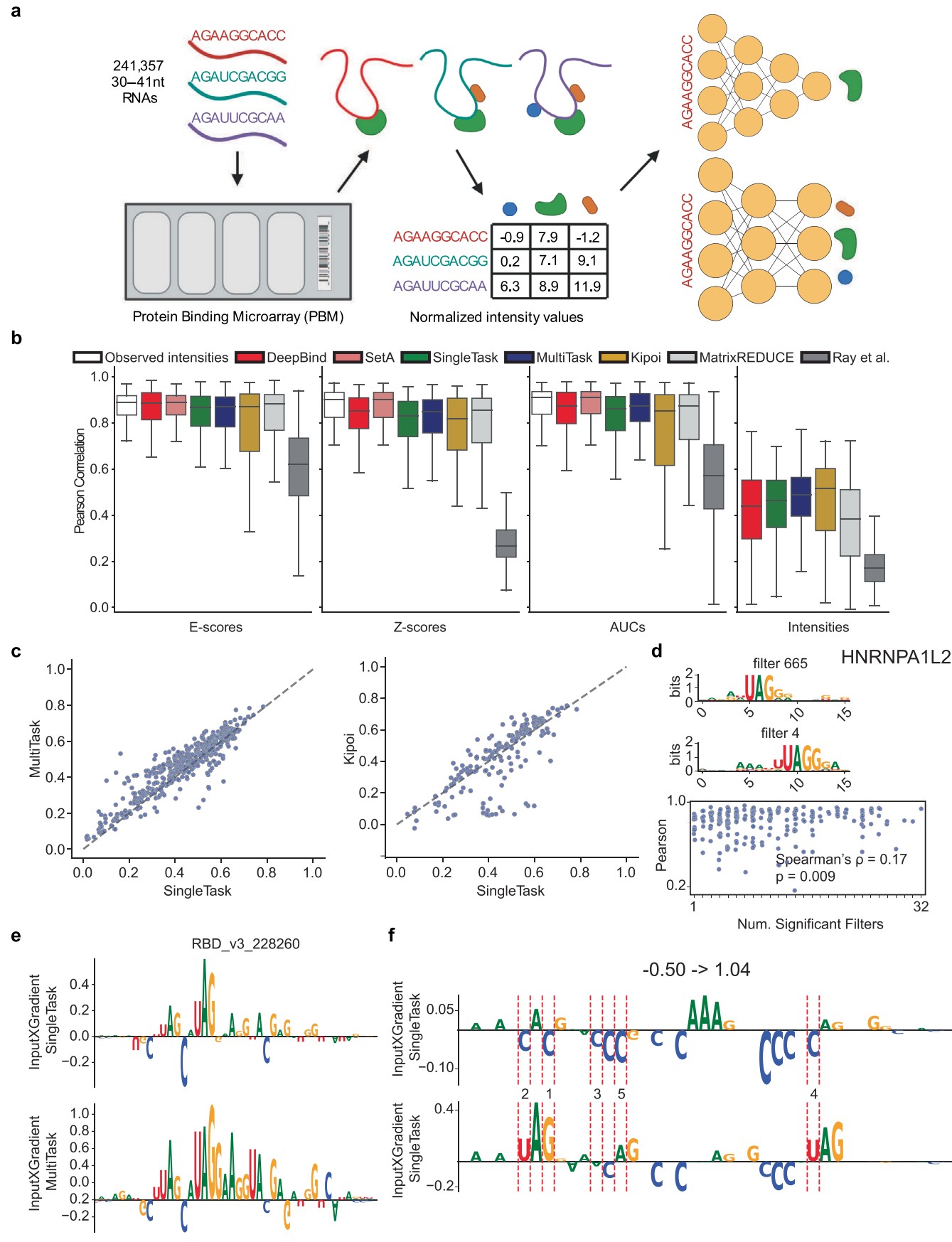

**Extended Data Fig. 1 | See next page for caption.**

**Extended Data Fig. 1 | In vitro RNA binding prediction with DeepBind. a**, ray13 use case schematic. n = 244 RNA binding proteins (RBPs) were assayed across a set of 241,357 RNA probes to generate a 241,357 × 244 dimensional matrix of normalized intensity values. **b**, Pearson correlations across four different metrics with each metric calculated from comparisons between observed (Set B) and predicted binding intensities (see Methods for more details on how each metric is calculated). Each boxplot indicates a distribution of Pearson correlations across all n = 244 RBPs, except for Kipoi which includes n = 89 RBPs. Ray et al, MatrixREDUCE, DeepBind and Observed intensities refer to correlations calculated from predicted intensities reported in Alipanahi et al. Observed intensities and SetA refer to correlations calculated using the intensities from Set A probes as the predicted intensities (see Methods). The boxes show medians along with low and high quartiles. Whiskers extend to the furthest datapoint within 1.5-times the interquartile range. **c**, Performance comparison scatterplots for ST models against MT models (left) and against Kipoi models (right). Each dot indicates a comparison of the Pearson correlation between predicted and observed intensities for two models on a single RBP. **d**, (top) A multitask filter with a TomTom annotation for HNRNPA1L2 visualized as a PWM logo. (middle) A filter for the single task HNRNPA1L2 model with a TomTom annotation for HNRNPA1L2. (bottom) The relationship between multitask performance (using the Z-scored Pearson correlations of observed and predicted intensities) on the y-axis, against the number of filters that were annotated with the corresponding RBP for that task on the x-axis. The Spearman's correlation coefficient and associated *P*-value are shown. **e**, Attributions for the sequence with the highest observed intensity in the test set for HNRNPA1L2. The attributions were calculated using InputXGradient for single task (top) and multitask (bottom) models. **f**, The InputXGradient attribution scores for a random (top) and evolved (bottom) sequence after evolution with the HNRNPA1L2 single task model. Red dashed lines indicate mutations made during evolution and are annotated with the round the mutation occurred in. Source data.

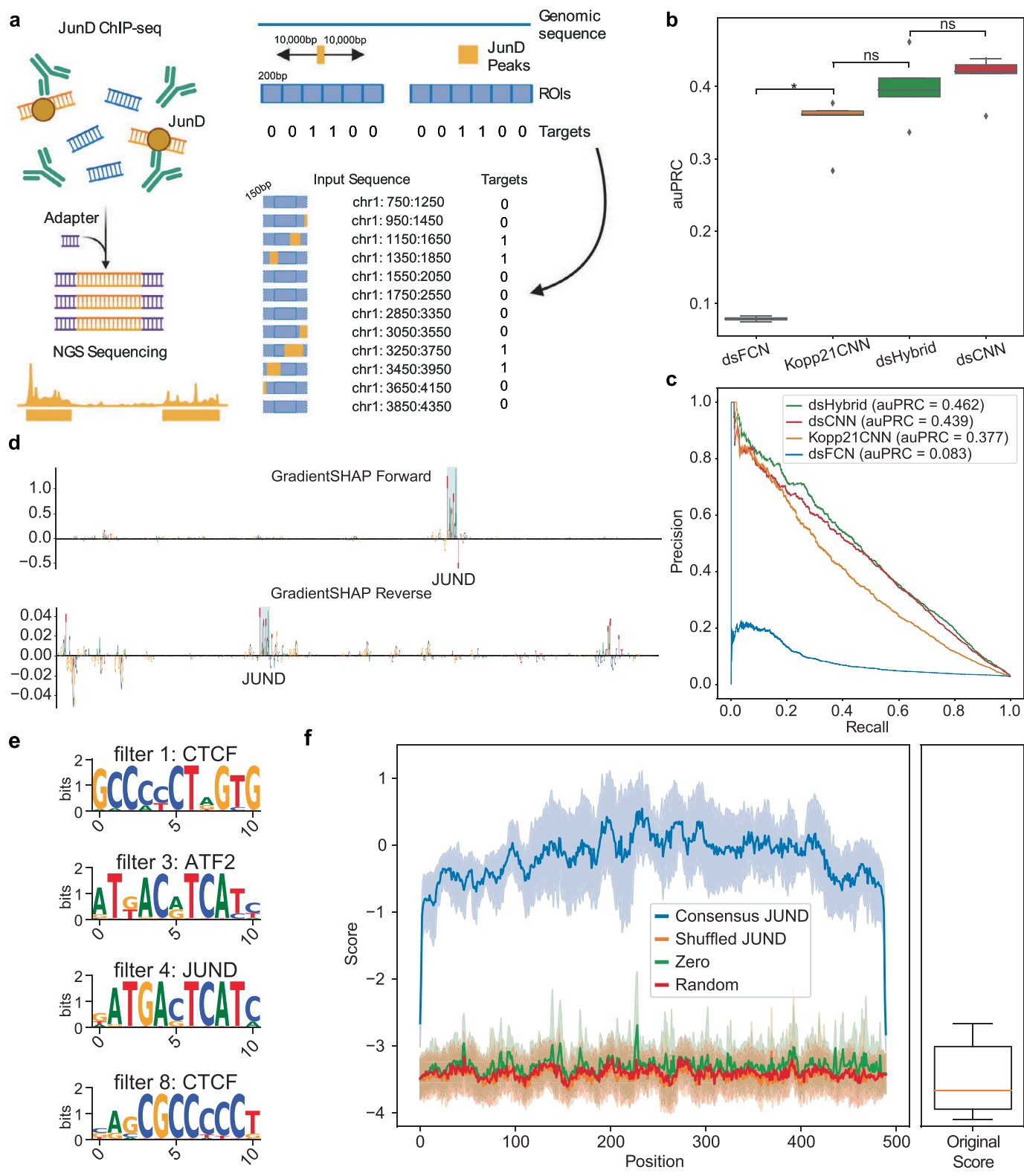

**Extended Data Fig. 2 | See next page for caption.**

**Extended Data Fig. 2 | JunD ChIP-seq binding classification. a**, kopp21 use case schematic. We used SeqData to load in a set of 11,086 ChIP-seq peaks for JunD and to generate positive and negative sets for JunD binding prediction. SeqData uses a set of regions of interest (ROIs) along with peaks and a bin size and outputs a set of labeled sequences for each bin in the ROI. Bins are labeled as positive (1) if they overlap a peak and negative (0) if they do not. Upon loading, each sequence is extended by 150 bp in each direction to provide more sequence context for prediction. **b,c**, auPRCs on held-out test data from chromosome 3 for JunD binding classification across four double-stranded architectures b, The boxplots show distributions of auPRC values on held-out test data for each architecture across n = 5 independent experiments (random initializations). A two-sided Mann-Whitney U test was used to determine $P$-values which were adjusted by the Benjamini-Hochberg method (* = adjusted $P$-value < 0.05, ns = not significant). Test statistics and adjusted $P$-values were: dsFCN-Kopp21CNN (u = 0, adjusted $P$-value = 0.02), dsFCN-dsHybrid (u = 0, adjusted $P$-value = 0.02), dsFCN-dsCNN (u = 0, adjusted $P$-value = 0.02), Kopp21CNN-dsHybrid (u = 4, adjusted $P$-value = 0.11), Kopp21CNN-dsCNN (u = 4, adjusted $P$-value = 0.11), dsHybrid-dsCNN (u = 8, adjusted $P$-value = 0.42). c, auPR curves for the best models from each architecture. **d**, Sequence logos of attributions for the top predicted sequence. The top row shows attributions from the forward strand and the bottom row from the reverse strand. Attributions were calculated using GradientSHAP. **e**, A selected set of convolutional filters visualized as PWM logos with significant annotations from TomTom (adjusted $P$-value < 0.05). **f**, Model scores for n = 10 random sequences with consensus JunD motif implanted at each possible location. Mean model scores with 95% confidence intervals are shown. The boxplot shows the distribution of scores for the random sequences prior to JunD motif implantation. All boxes show medians along with low and high quartiles. Whiskers extend to the furthest datapoint within 1.5-times the interquartile range. More extreme points are marked as outliers.

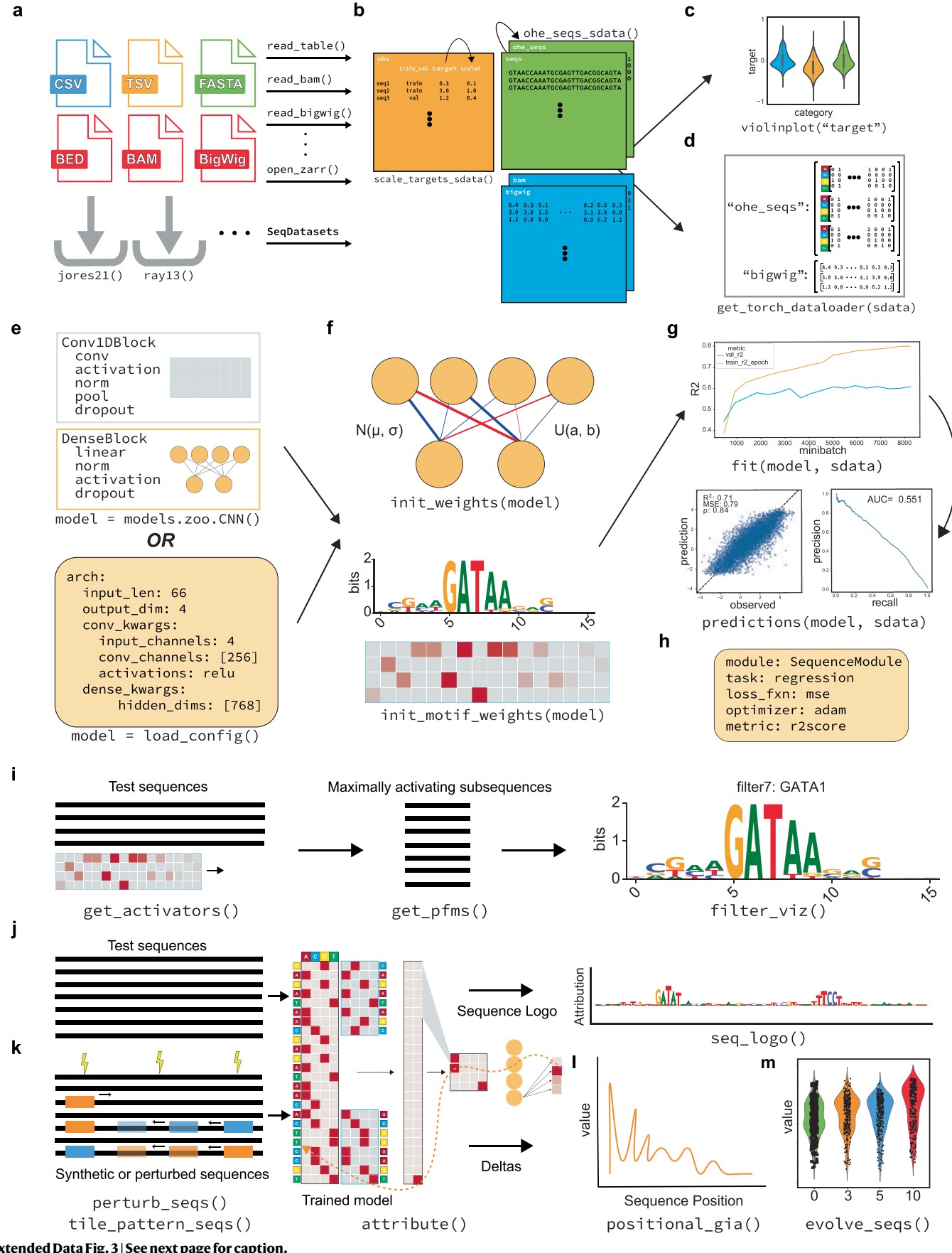

**Extended Data Fig. 3 | See next page for caption.**

**Extended Data Fig. 3 | End-to-end data processing, training, and interpretation with EUGENe. a**, SeqData objects can be loaded from files already on disk, or by calling for a dataset available for download from the SeqDatasets subpackage. Once instantiated, SeqData objects containerize the EUGENe workflow, easing the **b**, preprocessing of sequences and of sequence metadata, **c**, the generation of exploratory data analysis plots, and **d**, the creation of PyTorch loadable datasets and objects. **e**, An architecture can be instantiated either from a single function call (top) or from a configuration file (bottom). Conv1DBlocks and DenseBlocks both allow for flexibility in the ordering of layers they contain (one example ordering is shown). Instantiated architectures can be **f**, initialized with a desired initialization scheme, then **g**, fit to training data and used to predict on held-out data. Performance metric training curves are pictured in the top panel of **g**, test set performance curves for regression (left) and classification (right) are depicted in the bottom panel of **g**. Both training

and prediction are handled by PyTorch Lightning. We show an example of the arguments for instantiating a SequenceModule in **h**. **i**, For filter interpretation, filters in the first convolutional layer are used to scan input sequences for 'maximally activating subsequences' that can then be used to generate position frequency matrices and sequence logos. **j**, Attribution analysis starts by passing inputs sequences through the model to generate an output. This output signal is then backpropagated to the input to generate a per nucleotide score that can be visualized as a sequence logo. **k**, Random or synthetically designed sequences that have been mutated or have had motifs implanted in them can be scored using a trained model. Results from toy examples of this in silico approach are shown in **l** and **m**, which depict a positional importance analysis and a prediction evolution analysis respectively. Example function arguments have been omitted for a, b, e, i, j, k, l and m.

# Reporting Summary

## Statistics

For all statistical analyses, confirm that the following items are present in the figure legend, table legend, main text, or Methods section.

| n/a | Confirmed | |
|---|---|---|
| ☐ | ☒ | The exact sample size (*n*) for each experimental group/condition, given as a discrete number and unit of measurement |
| ☐ | ☒ | A statement on whether measurements were taken from distinct samples or whether the same sample was measured repeatedly |
| ☐ | ☒ | The statistical test(s) used AND whether they are one- or two-sided *Only common tests should be described solely by name; describe more complex techniques in the Methods section.* |
| ☐ | ☒ | A description of all covariates tested |
| ☐ | ☒ | A description of any assumptions or corrections, such as tests of normality and adjustment for multiple comparisons |
| ☐ | ☒ | A full description of the statistical parameters including central tendency (e.g. means) or other basic estimates (e.g. regression coefficient) AND variation (e.g. standard deviation) or associated estimates of uncertainty (e.g. confidence intervals) |
| ☐ | ☒ | For null hypothesis testing, the test statistic (e.g. *F*, *t*, *r*) with confidence intervals, effect sizes, degrees of freedom and *P* value noted *Give P values as exact values whenever suitable.* |
| ☒ | ☐ | For Bayesian analysis, information on the choice of priors and Markov chain Monte Carlo settings |
| ☒ | ☐ | For hierarchical and complex designs, identification of the appropriate level for tests and full reporting of outcomes |
| ☐ | ☒ | Estimates of effect sizes (e.g. Cohen's *d*, Pearson's *r*), indicating how they were calculated |

*Our web collection on statistics for biologists contains articles on many of the points above.*

## Software and code

Policy information about availability of computer code

| Data collection | All datasets used in this study are publicly available and were collected from the sources listed in their respective publications. |
|---|---|
| Data analysis | The EUGENe source code is available on GitHub at https://github.com/ML4GLand/EUGENe and as a package on PyPi at https://pypi.org/project/eugene-tools/. Code that utilizes the package to produce the results presented in the manuscript can be found on GitHub at https://github.com/ML4GLand/EUGENe_paper. For the results presented and discussed in the manuscript, we used EUGENe v0.1.2. Other software versions for packages mentioned in the manuscript include:<br><br>Captum v0.5.0<br>Dask v2023.3.24<br>MotifData v0.0.1<br>PyTorch Lightning v2.0.0<br>SeqData v0.0.1<br>SeqDatasets v0.0.1<br>SeqExplainer v0.0.1<br>SeqPro  v0.1.3<br>PyTorch v2.0.0<br>Xarray v2023.4.0<br>Yuzu v4.1.1<br>Zarr v2.14.2<br>modisco-lite v2.0.7<br>bpnet-lite v0.5.1 |

```
bedtools v2.31.0
Kipoi v0.8.6
ushuffle v1.1.2
RayTune 2.4.0
Seaborn v0.12.2
Matplotlib v3.6.2
Numba v0.57.0
NumPy v1.23.5
Pandas v1.5.2
TomTom v5.5.4
LogoMaker v0.0.8
```

For manuscripts utilizing custom algorithms or software that are central to the research but not yet described in published literature, software must be made available to editors and reviewers. We strongly encourage code deposition in a community repository (e.g. GitHub). See the Nature Portfolio guidelines for submitting code & software for further information.

## Data

Policy information about availability of data

All manuscripts must include a data availability statement. This statement should provide the following information, where applicable:
- Accession codes, unique identifiers, or web links for publicly available datasets
- A description of any restrictions on data availability
- For clinical datasets or third party data, please ensure that the statement adheres to our policy

All datasets used in this study are publicly available. Raw and processed data for the plant promoter STARR-seq were obtained from https://github.com/tobjores/Synthetic-Promoter-Designs-Enabled-by-a-Comprehensive-Analysis-of-Plant-Core-Promoters. Normalized RNA probe binding intensities were obtained from http://hugheslab.ccbr.utoronto.ca/supplementary-data/RNAcompete_eukarya/norm_data.txt.gz. JunD peaks from human embryonic stem cells (H1-hesc) called with the hg38 reference genome were obtained from encodeproject.org (ENCFF446WOD, conservative IDR thresholded peaks, narrowPeak format). Blacklisted regions for hg38 were obtained from http://mitra.stanford.edu/kundaje/akundaje/release/blacklists/hg38-human/hg38.blacklist.bed.gz. TomTom queries were performed against the Ray2013 Homo sapiens and the HOCOMOCO v11 FULL motif collections for the RBP binding and JunD binding use cases respectively. The JunD PFM was obtained from https://jaspar.genereg.net/matrix/MA0491.1 for the in silico implantation experiment. 89 RDP models were obtained from the Kipoi mode repository at https://kipoi.org/models/DeepBind/Homo_sapiens/RBP/. We have also deposited the EUGENe specific dataset files and trained models used in the analyses presented here on Zenodo (Klie 2023). These represent the processed data files and SeqData objects that can be used along with the accompanying code to generate the figures for all the use cases. Source data for Figure 2, Extended Data Figure 1, and Extended Data Figure 2 is available with this manuscript.

## Human research participants

Policy information about studies involving human research participants and Sex and Gender in Research.

| Reporting on sex and gender | not applicable |
|---|---|
| Population characteristics | not applicable |
| Recruitment | not applicable |
| Ethics oversight | not applicable |

Note that full information on the approval of the study protocol must also be provided in the manuscript.

# Field-specific reporting

Please select the one below that is the best fit for your research. If you are not sure, read the appropriate sections before making your selection.

☒ Life sciences  ☐ Behavioural & social sciences  ☐ Ecological, evolutionary & environmental sciences

For a reference copy of the document with all sections, see nature.com/documents/nr-reporting-summary-flat.pdf

# Life sciences study design

All studies must disclose on these points even when the disclosure is negative.

| Sample size | The three datasets used in the manuscript represent diverse data types and modeling tasks that highlight core aspects of EUGENe and showcase its flexibility. We replicated analyses with the same sample sizes as previously reported, which were sufficient to train complex machine learning models for analysis of these datasets |
|---|---|
| Data exclusions | No data were excluded from these analyses. |

| Replication | All models were trained across 5 random initializations to assess reproducibility. We observed successful replication across the 5 trials in all three datasets. |
| Randomization | Training, validation, and test sets for training models were generated via random splits of the full datasets or as described in previous publications when applicable. |
| Blinding | Investigators were not blinded to group allocation during data analysis as knowledge of group allocation was necessary for training models. |

# Reporting for specific materials, systems and methods

We require information from authors about some types of materials, experimental systems and methods used in many studies. Here, indicate whether each material, system or method listed is relevant to your study. If you are not sure if a list item applies to your research, read the appropriate section before selecting a response.

## Materials & experimental systems

| n/a | Involved in the study |
|-----|------------------------|
| ☒ ☐ | Antibodies |
| ☒ ☐ | Eukaryotic cell lines |
| ☒ ☐ | Palaeontology and archaeology |
| ☒ ☐ | Animals and other organisms |
| ☒ ☐ | Clinical data |
| ☒ ☐ | Dual use research of concern |

## Methods

| n/a | Involved in the study |
|-----|------------------------|
| ☒ ☐ | ChIP-seq |
| ☒ ☐ | Flow cytometry |
| ☒ ☐ | MRI-based neuroimaging |

