## [Peer Review File · Nature Computational Science]

Peer Review Information

Journal: Nature Computational Science

Manuscript Title: Predictive analyses of regulatory sequences with EUGENE

Corresponding author name(s): Professor Hannah Carter

Editorial Notes:

Transferred manuscripts This manuscript has been previously reviewed at another journal that is not operating a transparent peer review scheme. This document only contains reviewer comments, rebuttal and decision letters for versions considered at Nature

Reviewer Comments & Decisions:

Decision Letter, initial version:

Date: 3rd March 23 08:52:17

Last Sent: 3rd March 23 08:52:17

Triggered By: Fernando Chirigati

From: fernando.chirigati@us.nature.com

To: hkcarter@ucsd.edu

Subject: Decision on Nature Computational Science manuscript NATCOMPUTSCI-23-0044-T

Message: Dear Professor Carter,

Your manuscript "EUGENE: A Python toolkit for predictive analyses of regulatory sequences" has now been seen by 3 referees, whose comments appear below. In the light of their advice, we have decided that we cannot offer to publish your manuscript in Nature Computational Science.

From the reports, you will see that while they find your work of some interest and importance, the referees raise concerns about the potential limited novelty and practical

usefulness of the proposed tool. We feel that these criticisms are sufficiently important as to preclude publication of your work in Nature Computational Science.

I am sorry that we cannot be more positive on this occasion, but hope that you find the referees' comments helpful when preparing your paper for resubmission elsewhere.

Best,
Fernando

--

Fernando Chirigati, PhD
Chief Editor, Nature Computational Science
Nature Portfolio

Reviewers' Comments:

Reviewer #1 (Remarks to the Author):

The authors propose EUGENE, a FAIR toolkit for the analysis of labeled sets of nucleotide sequences using deep learning. The toolkit is built using PyTorch and includes modules for data extraction, transformation, loading from various file formats, instantiation, initialization, and training of different model architectures, as well as model evaluation and interpretation. According to the authors, the toolkit is designed to be simple, modular, and extensible and has been applied to three previously studied predictive modeling tasks in genomics: plant promoter activity prediction, RNA binding prediction, and transcription factor binding classification. The toolkit can be used to develop and deploy workflows on new or existing datasets, retrain seminal model architectures, or import models from Kipoi.

The idea that EUGENE is proposing is not completely novel. There has been a previous python package Janguu¹ that also proposes a data loading format for genomics data and interaction with deep learning libraries which the authors also state. In fact, some of EUGENE's dataloaders are wrapper functions to Janguu. The novelty compared to Janguu is the implementation of PyTorch models and the implementation of model interpretation designed for genomic sequences. Eugene could be very useful for deep learning in genomic sequences but this is dependent on the level of maintenance the authors will put into the code. In particular, the out-of-memory training that is offered by similar tools^{1,2} could be highly useful to train large-scale models as well as large single-cell datasets.

In summary, I recommend a major revision.

Major points

=====

1. It is unclear how useful this tool will be with respect to the listed limitation, especially since everything has to be loaded to memory. It would be useful to have some performance statistics of how long the sequences and the dataset can be.
2. The documentation is not always complete and does not include all parameters. For example, Cnn_kwargs is unfortunately not defined in the CNN model. The same holds true

for tasks where the different options are not defined. It also seems that some model combinations have not been tested. For instance, the CNN model does not work with the option 'binary_classification'

3. It seems like the software is mainly developed to work with one or multiple target values per sequence but not with profile data which is often used in genomics models (BPNet, Informer, Bassenji etc.) e.g. the ChIP-seq profile at each position or binned tracks as for Enformer. It is unclear how this could be achieved with SeqData and Eugene.

4. The method only suggests very simplistic train test splits of sequence. But especially in genomics there are often more complicated scenarios. For example in multi-species training, one needs to split by homologues. It would be very useful to have this implemented or at least discussed.

Minor points

=====

5. The tutorial fails, as there is an error in the definition of the paths. The tutorial should either be moved from /docs to /tutorial, or paths in the notebook should be changed:

```
eu.settings.config_dir = "./tutorial_configs"
```

```
-> eu.settings.config_dir = "../tutorials//tutorial_configs"
```

6. The dataloader does not differentiate between csv and tsv files

7. The eu.dl.read_bam documentation talks about bed files

Reviewer #1 (Remarks on code availability):

See above.

Reviewer #2 (Remarks to the Author):

This work introduces FAIR software comprised of base classes and wrapper functions for simplifying the analysis of genomic deep learning in Pytorch. The authors first describe the key functionalities of the package (data preparation, model definition, evaluation and interpretation), the two central object classes in the package (SeqData, BaseModel) and 3 use cases of the package to work with published datasets and models. The presentation is well written and the accompanying code is very readable as well.

Although the authors have done a great job at setting the foundations and mention in the paper that the package is extensible (and that they intend to add more functions), some key functionalities are missing that are widely used in SOTA models. Also, better intuition for how to use the model interpretability tools demonstrated in the paper could be useful to prevent empowering users to over-interpret their data.

This software has potential. However, in its current form, it has a feel of being a bit dated in terms of models and analyses. Here are some suggestions that could improve its utility.

Concerns:

1. Since if someone was only going to use one aspect of EUGENE, they could just use the tools themselves (i.e. janggu, captum, pytorch lightning, logomaker), the utility of EUGENE is that you can do a more streamlined end-to-end analysis. So, in addition to highlighting the functionality and flexibility, it might make sense to list what end-to-end analysis is available. It is not clear what it currently can do and what is suggested that can be incorporated with customization. For instance, is it able to generate multi-task binary classification datasets from a set of bed files? is it able to generate base-resolution regression task from bigwigs? Or is it only limited to generating single-task binary classification of ChIP-seq bed files? Are models only capable of scalar predictions for each task?
2. Expanded motif analysis. Motif analysis could in principle be accomplished with off-the-shelf motif analysis tools that have been well established in the bioinformatics community. A major benefit of deep learning is that it can capture a richer set of information, such as motif interactions and sequence context. Therefore, it would be beneficial to include interpretability tools that go beyond standard motif analysis, eg. distance-dependent motif cooperativity by embedding multiple motifs across background sequences used in DeepSTARR, BPNNet and several others. It seems that the current framework could easily support this functionality.
3. Memory limitations. A major concern is that the entire dataset has to be loaded into memory. For many deep learning applications, this is not feasible even for DGX systems. PyTorch supports reading chunks of data during training and this should be easily included in EUGENE. Otherwise, this would be a major shortcoming of the package and this should be made more clear upfront within the scope of the paper (i.e. abstract and introduction).
4. Base-resolution predictive modeling. It is unclear what tools are available for working with genome-wide regression models other than reading in BigWig files. Such models, like Bassenji, BPNNet, Enformer, among others, are becoming increasingly popular. Thus, providing access to these base (or binned) resolution regression analysis would be a major contribution. While the other analyses (Figures 3-5) are quite basic (with pytorchlightning doing most of the heavy lifting), they do still comprise the most popular use cases of deep learning in this space.
5. Hyperparameter search. Another major limitation is the lack of support for hyperparameter optimization. Models that are optimized for one dataset typically benefit from fine-tuning when analyzing another, smaller dataset. What functionality is there in EUGENE to facilitate this process? Weights&Biases could aid this process.
6. Fine-tuning pre-trained models. There was no mention of how a pretrained model can be fine-tuned using EUGENE. Could the weights of the trunk be fixed? Could the new output layer and the parameters in the trunk be trained with different learning rates?
7. Filter interpretability can be meaningful or not. Thus a simple cursory visualization and a TomTom search may lead to erroneous interpretations of important motifs. It is well known that Tomtom is quite inaccurate when performing a motif search with convolutional filters (Ullah & Ben-Hur, NAR, 2021, among many others). Some thoughtful guidance should be provided to promote robust science and prevent over-interpretation.
8. Limitations of attribution analysis. Several of the attribution maps shown in this paper also seem quite noisy. This is well known that attribution maps are sensitive to local

function properties and thus they may or may not be reliable. (Han et al. "Which explanation should i choose? a function approximation perspective to characterizing post hoc explanations." arXiv (2022).) Again, a thoughtful explanation could help users understand that while EUGENE provides functionality for attribution analysis, this may or may not constitute a biologically meaningful interpretation.

9. Analysis of prediction evolution. This analysis is quite straightforward to evolve sequences 1 mutation at a time to greedily increase predictions. But it's not clear what insights could be gained from that analysis. There should be a more thoughtful discussion on the purpose of this analysis and how to study the properties of the mutated sequences to facilitate "interpretation" of the key factors that increase predictions. I appreciate this is a software tool and not biology paper, but demonstrating functionality should include why one would do it and what information one could gain from such an analysis.

10. The package is mainly suitable for complete beginners as more advanced users would be able to directly use the libraries (e.g. PytorchLightning, captum, logomaker) without needing the extra layer of wrapper functions. It should be made clear that the package is intended for novice users. If the desire is to engage existing researchers already in this space, then the functionality needs to incorporate more modern architectures and expanded model interpretability with in silico experiments.

11. I think it's worth thinking about what is the real limitation of wide-spread adoption of deep learning in genomics? Is it the difficulty of stringing together python packages (pytorch lightning, captum, logomaker)? Or is it a lack of understanding on how the models should be built and how to interpret the results from the filter analysis and attribution analysis?

Minor concerns:

1. SOTA models are not up to date - DeepBind and DeepSEA are not SOTA models.
2. what is flexibility of data processing? Is it straightforward to build datasets with other customizations? Eg. using DNase-seq peaks as negative examples, which is typically done. Can the models easily incorporate additional features, such as epigenetic tracks, eg. DNase-seq, as is done for several winning models in the ENCODE-DREAM challenge?
3. The order of the basic conv module is conv, activation, pool (optional), dropout (optional), batch norm (optional). From model to model, the order of these can be employed differently, eg. Basset, DeepSTARR.
4. Should cite captum as sources for attribution maps.

Reviewer #2 (Remarks on code availability):

The code is organized well. I did not try to install or run the code.

Reviewer #3 (Remarks to the Author):

In this paper, Kile et al. presented a python package, EUGENE, for training genomic sequence deep learning models. The package is based on PyTorch, a popular deep learning

framework. Compared to existing toolkits for sequence-based deep learning software, EUGENE is designed to be simple and extensible. The authors then applied the package to three prediction tasks published previously.

Although I appreciate the non-trivial software development efforts from the author, I have concerns regarding the novelty of this work, and how it positions itself for novice and expert ML users. The use cases are over-simplified. My comments are below.

1. The EUGENE package presents two python class objects SeqData and BaseModel. This requires considerable knowledge of Python object-oriented programming, which is relatively less user-friendly than text configuration-based framework, such as Selene. Therefore, it poses a barrier to novice ML users.
2. For users who are expert in python and/or ML, what's the advantage of using EUGENE's `BaseModel` class compared to pytorch-lightning's `LightningModule`?
3. The functionality and API of EUGENE is highly similar to Janggu. Although Janggu is based on Keras, while EUGENE is based on PyTorch, this backend difference does not seem scientifically significant to me.
4. The input sequence length in the case studies are 170bp, 200bp, 200bp, for each of the three use cases, respectively. They are substantially smaller than current powerful models, such as Enformer (200kb), Sei (4096bp), or even earlier works such as Basenji (131kb) and DeepSEA (1000bp). The authors should consider tasks with enlarged input sequence size to match the current standards.
5. To show real-world application, the authors should also consider more complex architectures that have been shown effective in genomics recently, such as residual connections, inception layers, and attention layers.
6. As a high-level software package, it is also important to show if a model built by EUGENE can train as efficiently as in PyTorch-lightning and in native PyTorch, such as GPU utilization rate, GPU time.

** Although we cannot publish your paper, it may be appropriate for another journal in the Nature Portfolio. If you wish to explore the journals and transfer your manuscript please use our <https://mts-natcomputsci.nature.com/cgi-bin/main.plex?el=A3DI1BNA5A2HCM5X4A9ftd63wDxLQyX5UJTIVWxO9pNgZ> manuscript transfer portal. You will not have to re-supply manuscript metadata and files, but please note that this link can only be used once and remains active until used. For more information, please see our http://www.nature.com/authors/author_resources/transfer_manuscripts.html?WT.mc_id=EMI_NPG_1511_AUTHORTRANSF&WT.ec_id=AUTHOR manuscript transfer FAQ page.

Note that any decision to opt in to In Review at the original journal is not sent to the receiving journal on transfer. You can opt in to *[In Review](https://www.nature.com/nature-research/for-authors/in-review)* at receiving journals that support this service by choosing to modify your manuscript on transfer. In Review is available for primary research manuscript types only.

For Nature Portfolio general information and news for authors, see <http://npg.nature.com/authors>

Author Rebuttal to Initial comments

Summary of responses

We thank the reviewers for the many helpful suggestions that have strengthened our manuscript. We have summarized the common themes brought up by reviewers and how we have addressed them below. Following the summary, we have included point-by-point responses to each individual reviewer. **Revised text included in the manuscript is indicated in dark blue, reviewer comments in black, and responses and old manuscript text in light blue**

1. Memory efficiency

Reviewer's 1 and 2 expressed concerns with EUGENE's memory efficiency. We have revamped EUGENE's data loading capabilities to now allow for completely out-of-core training. We implemented this in a new standalone Python subpackage called SeqData that includes all the previous functionality offered without the need to load data into CPU memory. During training, EUGENE always processes datasets in mini-batches and *does not* require that the whole dataset fit into GPU memory, regardless of whether or not the dataset is loaded into CPU memory. We have included a new section of the manuscript dedicated to discussing EUGENE's new data loading capabilities with SeqData that includes a formal analysis of EUGENE's memory footprint in **Supplementary Figure 1**.

2. Data types supported

All three reviewers asked for a clearer explanation of the types of datasets that EUGENE supports. Of note, they asked whether EUGENE was capable of handling longer sequence lengths than presented in the manuscript, and whether modeling binned and base-pair resolution datasets was possible. Longer sequence lengths are addressed in the memory footprint analysis mentioned above (**Supplementary Figure 1**). In short, the longest sequence length is roughly constrained to whether a single sequence fits into CPU and GPU memory. Modeling binned and base-pair resolution data is also supported and has now been made completely native to EUGENE with SeqData (i.e. Janggu is no longer a requirement). We address the reviewers' concerns about the specific data types supported in our point-by-point responses and have added manuscript text that more explicitly details EUGENE's capabilities. This includes a description of the types of datasets that can be constructed with SeqData (see the new **Data extraction, transformation and loading with SeqData** section) and the types of end-to-end analyses supported with EUGENE (see the first paragraph in the **EUGENE use cases** section and **Table 1**). We have also added a section to the online documentation dedicated to listing the different end-to-end workflows that are possible in EUGENE and added online notebooks that demonstrate how to work with base-pair resolution datasets (with links to both in the manuscript).

3. Utility for beginners vs advanced users

Reviewers 2 and 3 expressed concerns with how EUGENE aligns itself to advanced users and beginners. We designed EUGENE in a way that benefits both, but the primary goal of

the package is to reduce accessibility barriers to common end-to-end workflows. We have modified the manuscript in several ways to better reflect this goal, including in the section dedicated to a description of EUGENE's utility in practice (see the first paragraph of the **EUGENE use cases** section and **Table 1**).

With respect to novice users, EUGENE is an excellent place to get started with deep learning for regulatory genomics. It makes the entire workflow available via a single installation and provides an intuitive interface for those familiar with Python, and extensive documentation for those who are less so. Reviewer 2 brought up important points about the ease at which a novice user might over-interpret results from running model interpretation methods. We reorganized our presentation of model interpretation into a dedicated section (see **Model interpretation with SeqExplainer**) that now addresses guidance on the usage and limitations in interpreting filters, attributions and *in silico* evolution. We have also added sections and tutorials to EUGENE's documentation that are dedicated to model interpretation and avoiding over-interpretation.

We have implemented several additions to EUGENE that are aimed at making the toolkit more useful for advanced users. This includes separating EUGENE's functionality into standalone subpackages, such that advanced users can install and use just what they need, and providing modern model architectures such as BPNet (see **Supplementary Table 3**). EUGENE also adds functionality complementary to available tools (e.g. functions for instantiating convolutional filters from MEME files), reduces the need for repeated use of boilerplate code (e.g. training loops and model definitions), and mitigates the opportunity for human errors to arise in bespoke implementations of methods.

4. Requests for additional functionality

Each reviewer also had suggestions for additional functionality. Reviewer 1 suggested an expanded set of train-test splitting functions, reviewer 2 called for an expanded motif analysis suite and hyperparameter optimization support, and reviewer 3 suggested more complex layers be made available. In response, we have implemented homology based train-test splitting, distance-dependent motif cooperativity analysis, hyperparameter optimization support, and residual, inception and attention layers in EUGENE (these have all been removed as future directions in the discussion). For all these additions, we have added tutorial notebooks available on GitHub (<https://github.com/ML4GLand/tutorials/tree/main/eugene>). We also believe that many of these suggestions are excellent candidates for community based development and have designed EUGENE to accommodate this. For example, we designed the SeqExplainer subpackage to provide flexible building blocks for performing a multitude of different motif analyses and anticipate continuing to add high-level functions for streamlining several common analyses to EUGENE in the future.

5. Updates to manuscript organization

To accommodate the changes made to the codebase and to our responses to the reviewer's comments, we have made several modifications to the manuscript's organization. The manuscript now has only two main **Results** sections (previously three). The first, titled **The EUGENE workflow**, contains three subsections, one for each major stage of the workflow. **Figure 2** has been modified to accommodate this, and now includes the content from the old **Supplementary Figure 1**. The new second section, titled **EUGENE use cases**, also has three subsections (one for each use case) as it previously did, but now includes a brief description of the types of end-to-end analyses supported by EUGENE in its first paragraph (along with **Table 1**). Finally, updates we have made to the code have necessitated that we retrain models with the latest version of the package. This led to small discrepancies in the results that are reflected in updates to **Figures 3, 4, and 5** (and **Supplementary Figures 3, 4 and 5**). These did not, however, change any of the findings previously reported.

Reviewer #1

Remarks to the Author

The authors propose EUGENE, a FAIR toolkit for the analysis of labeled sets of nucleotide sequences using deep learning. The toolkit is built using PyTorch and includes modules for data extraction, transformation, loading from various file formats, instantiation, initialization, and training of different model architectures, as well as model evaluation and interpretation. According to the authors, the toolkit is designed to be simple, modular, and extensible and has been applied to three previously studied predictive modeling tasks in genomics: plant promoter activity prediction, RNA binding prediction, and transcription factor binding classification. The toolkit can be used to develop and deploy workflows on new or existing datasets, retrain seminal model architectures, or import models from Kipoi.

The idea that EUGENE is proposing is not completely novel. There has been a previous python package Janguu¹ that also proposes a data loading format for genomics data and interaction with deep learning libraries which the authors also state. In fact, some of EUGENE's dataloaders are wrapper functions to Janguu. The novelty compared to Janguu is the implementation of PyTorch models and the implementation of model interpretation designed for genomic sequences. Eugene could be very useful for deep learning in genomic sequences but this is dependent on the level of maintenance the authors will put into the code. In particular, the out-of-core training that is offered by similar tools^{1,2} could be highly useful to train large-scale models as well as large single-cell datasets.

In summary, I recommend a major revision.

Major points

1. It is unclear how useful this tool will be with respect to the listed limitation, especially since everything has to be loaded to memory. It would be useful to have some performance statistics of how long the sequences and the dataset can be.

We agree with the reviewer that out-of-core training is an important piece of functionality for EUGENE to include. We have updated EUGENE's data loading format to address this. SeqData no longer requires Janguu¹, but matches it's core functionality (listed at <https://janguu.readthedocs.io/en/latest/storage.html#genomic-datasets>):

- the ability to read from FASTA, BAM, BigWig and BED
- the ability to perform normalization of coverage and to specify custom coverage transformations
- the ability to specify the granularity of coverage
- a variety of data storage options and fast loading of datasets once written to disk
- the easy conversion to NumPy arrays, Pandas objects and PyTorch dataloaders
- the creation of dataset views

- the ability to perform all of these operations out-of-core

SeqData utilizes XArray² datasets loaded in as Dask³ arrays and backed by Zarr⁴ stores. The use of these packages rather than Janggu's GenomicArray and a GenomicIndexer object is beneficial for several reasons:

- 1) SeqData is more flexible and interactive than Janggu. Janggu requires that many preprocessing decisions be made prior to loading data in (e.g. the order of the one-hot encoding, sequence bin size, resolution of coverage, etc.). SeqData can do these operations out-of-core (see below) within a notebook interface. This is especially useful for performing exploratory data analyses and for model interpretation (which is often done on smaller subsets of the data).
- 2) SeqData aligns sequences, targets (e.g. read coverage) and metadata in a single object. This can be beneficial when applying the same transformations (e.g. simultaneous sequence and coverage trimming) to multiple variables and promotes a streamlined workflow.
- 3) The maintenance of the core data structures is handled by the large developmental teams of XArray, Dask and Zarr, all of which are NumFOCUS sponsored projects. As maintainers of SeqData, we can therefore focus on maintaining a smaller set of core functionality that utilizes these packages as they grow.

Most importantly, SeqData can now be used for completely out-of-core training. EUGENE's new effective memory limitation is the memory required to load in a single sequence. That is, as long as a single sequence can fit into both a user's CPU and GPU RAM, a EUGENE model can be trained on that dataset (in this case with a batch size of 1). In practice, training usually occurs in larger batches, but the principle remains the same.

To illustrate this, we generated datasets with increasing numbers of 10,000bp sequences and analyzed the peak memory usage required to load them on a machine with 2 CPU cores and 64GB of RAM (**Supplementary Figure 1a**, copied below). When we load the dataset into memory, as was done with the previous version of SeqData, we observe that peak memory usage increases until we reach 10M sequences, at which point we cross the 64GB RAM threshold and get an out-of-memory error. With the new out-of-core implementation, peak memory usage stays almost constant with the increasing number of sequences. GPU training is done using mini-batches regardless of whether the dataset is loaded into CPU memory, and there is never a requirement to store the whole dataset in memory on GPU.

Using this out-of-core functionality will, however, increase access time when processing and training on these datasets. For the same scenario described above, we see that iterating through batches takes about twice as long when done out-of-core (**Supplementary Figure 1b**, copied below). We also note that the access speed is dependent on other parameters that can lead to better results. Here we chose a chunk size (how many sequences Dask processes at a time) of 4096. We can see that this leads

to comparable performance with a 1000 sequence dataset loaded into memory. We have included this analysis as **Supplementary Figure 1** in the revised manuscript:

Supplementary Figure 1. Peak memory usage and batch processing time for datasets with increasing numbers of 10,000bp sequences. a, Peak memory usage against number of sequences for datasets loaded into memory versus datasets loaded out-of-core. **b,** Median time in seconds taken for processing a batch of 100 sequences using the same datasets as in **a**. Error bars indicate interquartile ranges across up to 100 random batches. Both axes in **a** and **b** are on the log₁₀ scale. All analyses were performed using a chunk size of 4096 along the sequence dimension on a machine with 2 CPU cores and 64GB of RAM.

As a user, it is useful to understand what resources and dataset size you are working with so you can maximize your data loading efficiency. Certain datasets, such as those used to train Enformer or Basenji, will likely require this out-of-core functionality. However, we have found that many useful and large datasets can entirely fit into memory on machines with modest CPU RAM (<32GB):

Dataset Name	Num Seqs	Sequence Length	In-memory size (GBs)	Num Tasks	URL
Cleaned Basset ENCODE training set	437478	600bp	~6GB	161 binary classification tasks	https://zenodo.org/record/7265991
base-pair resolution CTCF ChIP-seq training set	45803	2114bp	~4GB	2 signal tracks and 1 control track	https://github.com/kundajelab/baspairmodels
DeepSTARR training dataset	402296	249bp	~2GB	2 regression tasks	https://zenodo.org/record/5502060
Random promoter dataset	27991003	~130bp	~32GB	1 regression task	https://www.ncbi.nlm.nih.gov/geo/query/acc.cgi?acc=GSE104878

We have included this as **Supplementary Table 2** in the revised manuscript.

We have also included a new section of the manuscript dedicated to describing SeqData's updated specifications and its ability to perform out-of-core training (under the **Data extraction, transformation and loading with SeqData** section).

"By default, SeqData reads files from disk as XArray datasets⁴⁰ backed by Zarr stores⁵⁰ (**Figure 1a**). We chose to use XArray and Zarr as they are scalable, capable of handling high dimensional data, and have been previously utilized in a variety of bioinformatics domains⁵¹⁻⁵³. Furthermore, Zarr stores can be loaded out-of-core thanks to functionality offered by XArray and Dask⁵⁴, allowing for processing and training of large-scale datasets (**Supplementary Figure 1a**). As is standard in DL, training in EUGENE is always done by loading data into GPU memory in batches (when a GPU is available), but is slowed by using the out-of-core functionality on the CPU (**Supplementary Figure 1b**). Thus, the decision on whether to first load the dataset into CPU memory prior to training should balance available resources and dataset size. Certain datasets, such as those used to train Enformer²⁹ or Basenji¹⁸, will likely require this out-of-core functionality. However, we have found that many useful and large datasets can entirely fit into memory on machines with less than 32GB of RAM (**Supplementary Table 2**)."

(page 5, paragraph 2)

- The documentation is not always complete and does not include all parameters. For example, `Cnn_kwargs` is unfortunately not defined in the CNN model. The same holds true for tasks where the different options are not defined. It also seems that some model combinations have not been tested. For instance, the CNN model does not work with the option `'binary_classification'`

We thank the reviewer for pointing out these gaps in our documentation. We have addressed the missing `"cnn_kwargs"` argument, added more thorough documentation for the model instantiation functions (<https://eugene-tools.readthedocs.io/en/latest/api.html#models>), and added tutorial notebook dedicated to model instantiation (https://github.com/ML4GLand/tutorials/blob/main/eugene/models/instantiating_models.ipynb).

We could not replicate the issue mentioned with the CNN/"binary_classification" combination, but have performed more rigorous testing on this combination as well as others (<https://github.com/ML4GLand/EUGENE/blob/main/tests/notebooks/>).

- It seems like the software is mainly developed to work with one or multiple target values per sequence but not with profile data which is often used in genomics models (BPNet, Informer, Bassenji etc.) e.g. the ChIP-seq profile at each position or binned tracks as for Enformer. It is unclear how this could be achieved with SeqData and Eugene.

EUGENE is indeed capable of handling models that predict profile data. Single track or multiple tracks of data can be read into SeqData at base-pair resolution and binned if desired (we have a dedicated function for this operation that can keep the dataset out-of-core).

We showcase this functionality in a new example use case on EUGENE's "use cases" GitHub repository (https://github.com/ML4GLand/use_cases/tree/main/BPNet). We first downloaded a BPNet style dataset from stranded CTCF ChIP-seq and control tracks and preprocessed it according to <https://github.com/kundajelab/basepairmodels>. We then trained three BPNet models with three different pipelines. First, we used the `bpnet-lite` package (<https://github.com/jmschrei/bpnet-lite>) to load the data and train a model. Second, we use SeqData to load the data and `bpnet-lite` to train the model. Finally, we loaded the data with SeqData and trained a model with only EUGENE (all installed with EUGENE). Validation performance metrics for all models were similar:

	bpnet-lite only	SeqData + bpnet-lite	EUGENE only
Epoch	36	48	25
Iteration	51900	70000	37232
Validation MNLL	263.0577393	263.0949707	263.2419128
Validation Profile Pearson	0.4276593	0.42769808	0.42818016

Validation Count Pearson	0.73834264	0.73908126	0.73949087
Validation Count MSE	0.7036417127	0.6195657253	0.7754321098

To clarify EUGENE's capabilities, we have added the following paragraph (and accompanying **Table 1**) to the beginning of the **EUGENE use cases** section that discusses the end-to-end analyses that are possible with the toolkit:

"The major goal of EUGENE is to promote the effective design, implementation, validation and interpretation of DL solutions in regulatory genomics by streamlining end-to-end analyses. **Table 1** lists several common DL for regulatory genomics tasks that can be analyzed in an end-to-end fashion with EUGENE. We next describe three end-to-end analyses in detail, highlighting the core aspects of the workflow on different data types and training tasks. For links to code examples for other tasks described in **Table 1**, see EUGENE's associated "use cases" documentation page (https://github.com/ML4GLand/use_cases)." (page 9, paragraph 3)

Table 1. Common deep learning for regulatory genomics tasks can be run end-to-end with EUGENE

Task	Examples	Potential insights gained	ETL	Training and evaluation	End-to-end currently available?	Interpretation analyses currently available	Example in EUGENE use cases
Single task regression from a tabular file	DeepBind [10], ResidualBind [14]	Identification and quantification of motif importance on continuous or binary events (e.g. RBP binding)	Yes	Yes	Yes	Filter interpretation, attribution analysis, evolution, GIA	DeepBind
Single track classification of peak regions from a single bed file	DeepBind [10]	Identification and quantification of motif importance on binary events (e.g. TF binding)	Yes	Yes	Yes	Filter interpretation, attribution analysis, evolution, GIA	Kopp21
Multitask track classification (ChIP, ATAC, DNase, etc.) of peak regions from multiple bed files	DeepSEA [15], DanQ [16], Basset [17], Sei [23], Satori [26]	Identification and quantification of motif importance on biochemical activity (e.g. TF binding, transcription, DNA accessibility, etc.) Variant effects on biochemical activity	Yes	Yes	Yes	Filter interpretation, attribution analysis, evolution, GIA	Basset
Multitask track regression (ChIP, ATAC, DNase, etc.) at binned or base-pair resolution	Basenji [18], Enformer [29], BPNet [38]	Identification and quantification of motif importance on biochemical activity (e.g. transcription, DNA accessibility, etc.) Variant effects on biochemical activity CRE syntax rules	Yes	Yes	Yes	Filter interpretation, GIA	BPNet
Single task and multitask CRE activity prediction (both regression and classification (multiclass and multilabel))	DeepSTAR R [35], MPRA-DracoNN [36]	Identification and quantification of motif importance on CRE activity Variant effects on CRE activity CRE syntax rules	Yes	Yes	Yes	Filter interpretation, attribution analysis, evolution, GIA	DeepSTARR

Single cell ATAC-seq topic classification (multiclass classification)	DeepMEL [19], DeepMEL2 [20], DeepFlyBrain [24]	Identification and quantification of cell type specific motif importance Cell type specific variant effect prediction Cell type specific CRE syntax	Requires preprocessing with pycisTopic	Yes	Yes, with preprocessing performed by pycisTopic	Filter interpretation, attribution analysis, evolution, GIA	DeepMEL
Single cell ATAC-seq cell accessibility prediction*	scBasset [22]	Single cell analysis (denoising, imputation, clustering, etc.) Identification and quantification of cell type specific motif importance	Requires preprocessing with ScanPy	Yes	Yes, with preprocessing performed by ScanPy	Filter interpretation, attribution analysis, evolution, GIA	scBasset

4. The method only suggests very simplistic train test splits of sequence. But especially in genomics there are often more complicated scenarios. For example in multi-species training, one needs to split by homologues. It would be very useful to have this implemented or at least discussed.

We have integrated new functionality from GraphPart¹⁰ to now allow for train-test splitting based on the homology of sequences. Briefly, GraphPart first builds a weighted graph of pairwise similarities between each sequence and constructs partitions based on clusters in the graph. These partitions are then iteratively reassigned or removed until each partition has only sequences with homology below a user provided threshold. We have modified the following line in the **Data extraction, transformation and loading with SeqData** section of the manuscript to reflect this addition to the codebase:

"EUGENE includes a baseline set of functions for train and test set splitting (e.g. by chromosome, fraction, or homology⁶⁵) and target normalization (e.g. binning, z-score, clamping, etc.) (Figure 2b, left)." (page 5, paragraph 3)

There is substantial value in helping users avoid information leakage in training (and other machine learning pitfalls). We have added a notebook guide to the repository that is dedicated to illustrating how to perform proper train-test splitting (https://github.com/ML4GLand/tutorials/blob/main/eugene/preprocess/train_test_splits.ipynb). We have also mentioned this in the discussion:

"The use of bespoke methods for data preprocessing, as well as for model interpretation, is quite common in the field, and is often necessary to train accurate models that avoid common machine learning pitfalls⁶⁷. For example, some workflows may require complex implementations of train and test set splitting to protect against information leakage⁶⁸. We see substantial value in continuing to extend EUGENE into spaces such as these, and have designed the toolkit to allow for easy integration of this type of functionality. To continue to make bespoke methods and workflows accessible, we intend to encourage community development of EUGENE through tutorials, workshops and a dedicated user-group." (pages 18 and 19)

Minor points

5. The tutorial fails, as there is an error in the definition of the paths. The tutorial should either be moved from /docs to /tutorial, or paths in the notebook should be changed:

```
eu.settings.config_dir = "/tutorial_configs" ->  
eu.settings.config_dir = "./tutorials/tutorial_configs"
```

We thank the reviewer for pointing out this. We have fixed this issue in an updated tutorial notebook (https://eugene-tools.readthedocs.io/en/latest/basic_usage_tutorial.html)

6. The dataloader does not differentiate between csv and tsv files

The name of the function for reading tsv/csv files has been changed to `read_table`. Adding the `sep="\t"` argument to the function call allows tsv files to be read in. We did this to align EUGENE with how Pandas reads from csv.

7. The `eu.dl.read_bam` documentation talks about bed files

We thank the reviewer for identifying this inconsistency. We have updated SeqData's documentation (<https://seqdata.readthedocs.io/en/latest/api.html#default-readers>).

Remarks on code availability

See above.

Reviewer #2

Remarks to the Author

This work introduces FAIR software comprised of base classes and wrapper functions for simplifying the analysis of genomic deep learning in PyTorch. The authors first describe the key functionalities of the package (data preparation, model definition, evaluation and interpretation), the two central object classes in the package (SeqData, BaseModel) and 3 use cases of the package to work with published datasets and models. The presentation is well written and the accompanying code is very readable as well.

Although the authors have done a great job at setting the foundations and mention in the paper that the package is extensible (and that they intend to add more functions), some key functionalities are missing that are widely used in SOTA models. Also, better intuition for how to use the model interpretability tools demonstrated in the paper could be useful to prevent empowering users to over-interpret their data.

This software has potential. However, in its current form, it has a feel of being a bit dated in terms of models and analyses. Here are some suggestions that could improve its utility.

Major points

1. Since if someone was only going to use one aspect of EUGENE, they could just use the tools themselves (i.e. janggu, captum, pytorch lightning, logomaker), the utility of EUGENE is that you can do a more streamlined end-to-end analysis. So, in addition to highlighting the functionality and flexibility, it might make sense to list what end-to-end analysis is available. It is not clear what it currently can do and what is suggested that can be incorporated with customization. For instance, is it able to generate multi-task binary classification datasets from a set of bed files? Is it able to generate base-resolution regression tasks from bigwigs? Or is it only limited to generating single-task binary classification of ChIP-seq bed files? Are models only capable of scalar predictions for each task?

We thank the reviewer for this great suggestion. To provide a better overview of the diverse end-to-end analyses that are possible with EUGENE, we have modified the first paragraph of the **EUGENE use cases** section and added accompanying **Table 1**:

"The major goal of EUGENE is to promote the effective design, implementation, validation and interpretation of DL solutions in regulatory genomics by streamlining end-to-end analyses. **Table 1** lists several common DL for regulatory genomics tasks that can be analyzed in an end-to-end fashion with EUGENE. We next describe three end-to-end analyses in detail, highlighting the core aspects of the workflow on different data types and training tasks. For links to code examples for other tasks described in **Table 1**,

see EUGENE's associated "use cases" GitHub repository (https://github.com/ML4GLand/use_cases). (page 9, paragraph 3)

Table 1. Common deep learning for regulatory genomics tasks can be run end-to-end with EUGENE

Task	Examples	Potential insights gained	ETL	Training and evaluation	End-to-end currently available?	Interpretation analyses currently available	Example in EUGENE use cases
Single task regression from a tabular file	DeepBind [10], ResidualBin d [14]	Identification and quantification of motif importance on continuous or binary events (e.g. RBP binding)	Yes	Yes	Yes	Filter interpretation, attribution analysis, evolution, GIA	DeepBind
Single track classification of peak regions from a single bed file	DeepBind [10]	Identification and quantification of motif importance on binary events (e.g. TF binding)	Yes	Yes	Yes	Filter interpretation, attribution analysis, evolution, GIA	Kopp21
Multitask track classification (ChIP, ATAC, DNase, etc.) of peak regions from multiple bed files	DeepSEA [15], DanQ [16], Basset [17], Sei [23], Satori [26]	Identification and quantification of motif importance on biochemical activity (e.g. TF binding, transcription, DNA accessibility, etc.) Variant effects on biochemical activity	Yes	Yes	Yes	Filter interpretation, attribution analysis, evolution, GIA	Basset
Multitask track regression (ChIP, ATAC, DNase, etc.) at binned or base-pair resolution	Basenji [18], Enformer [29], BPNet [38]	Identification and quantification of motif importance on biochemical activity (e.g. transcription, DNA accessibility, etc.) Variant effects on biochemical activity CRE syntax rules	Yes	Yes	Yes	Filter interpretation, GIA	BPNet
Single task and multitask CRE activity prediction (both regression and classification (multiclass and multilabel))	DeepSTAR R [35], MPRA-DracoNN [36]	Identification and quantification of motif importance on CRE activity Variant effects on CRE activity CRE syntax rules	Yes	Yes	Yes	Filter interpretation, attribution analysis, evolution, GIA	DeepSTARR
Single cell ATAC-seq topic classification (multiclass classification)	DeepMEL [19], DeepMEL2 [20], DeepFlyBrain [24]	Identification and quantification of cell type specific motif importance Cell type specific variant effect prediction Cell type specific CRE syntax	Requires preprocessing with pycisTopic	Yes	Yes, with preprocessing performed by pycisTopic	Filter interpretation, attribution analysis, evolution, GIA	DeepMEL
Single cell ATAC-seq cell accessibility prediction*	scBasset [22]	Single cell analysis (denoising, imputation, clustering, etc.) Identification and quantification of cell type specific motif importance	Requires preprocessing with ScanPy	Yes	Yes, with preprocessing performed by ScanPy	Filter interpretation, attribution analysis, evolution, GIA	scBasset

We have also included a more detailed description of the types of datasets that can be loaded with SeqData, in the Data extraction, transformation and loading with SeqData section, and address the specific cases mentioned by the reviewer:

"The versatility of SeqData enables the generation of many custom datasets from combinations of these file types, including several commonly utilized in regulatory genomics. These include datasets derived from combinations of tabular and FASTA files that are suitable for single and multi-task regression and classification (e.g. DeepSTARR³⁵), datasets from genomic coordinates defined in BED files suitable for multi-task binary classification (e.g. DeepSEA¹⁵ or Sei²³), and datasets from multiple BigWigs and BED files suitable for binned or base-pair resolution regression (e.g. Basenji¹⁸ and BPNNet³⁸ respectively)." (pages 3 and 5)

We have also emphasized the importance of making end-to-end analyses accessible in several places:

"We designed EUGENE as a simple, flexible and extensible interface for streamlining several common end-to-end DL analyses, and illustrate these principles through application of the toolkit to three predictive modeling tasks." (Abstract, page 1)

"Generally, there is a need for an end-to-end toolkit in this space that follows FAIR data and software principles^{16,17} and that is inherently designed to be simple and extensible." (Introduction, page 3, paragraph 1)

"With EUGENE, we sought to integrate many of these aspects into an ecosystem of Python software that provides access to these workflows end-to-end." (Discussion, page 18, paragraph 2)

2. Expanded motif analysis. Motif analysis could in principle be accomplished with off-the-shelf motif analysis tools that have been well established in the bioinformatics community. A major benefit of deep learning is that it can capture a richer set of information, such as motif interactions and sequence context. Therefore, it would be beneficial to include interpretability tools that go beyond standard motif analysis, eg. distance-dependent motif cooperativity by embedding multiple motifs across background sequences used in DeepSTARR, BPNNet and several others. It seems that the current framework could easily support this functionality.

We agree with the reviewer that expanding the functionality in EUGENE for motif analysis would greatly improve a user's ability to extract more information from these deep learners. To meet this need, we developed the standalone SeqExplainer subpackage (included in the EUGENE installation) that builds on the model interpretation functionality included in the original submission. We introduce SeqExplainer in a new section of the manuscript titled Model interpretation with SeqExplainer:

"Interpreting models is a critical importance in regulatory genomics⁶⁵⁻⁶⁷, but is often made challenging by the complexity of neural networks and methods for their interpretation. To

address this in EUGENE, we created a standalone subpackage called SeqExplainer which makes various post-hoc interpretation strategies accessible to most PyTorch model trained on one-hot encoded genomic sequences (<https://github.com/ML4GLand/SeqExplainer>). SeqExplainer currently provides functionality for filter interpretation, attribution analysis, *in silico* experimentation, and sequence generation. Each strategy is briefly detailed below." (page 8, paragraph 2)

The subpackage includes two modules that enable a multitude of motif analyses: one dedicated to generating background sequences and the other dedicated to generating perturbations to these sequences (e.g. generating mutations, embedding motifs, occluding motifs, etc.). In EUGENE, we implemented wrapper functions for streamlining an analysis of a motif's positional importance (**Figures 3f** and **5f** in the manuscript) and the suggested distance-dependent motif cooperativity analysis. In a separate use case (https://github.com/ML4GLand/use_cases/tree/main/DeepSTARR), we use the latter functionality to investigate distance-dependent motif cooperativity with the DeepSTARR model trained in ²¹. We will continue to add wrappers for these types of functions to SeqExplainer, but also believe that these types of analyses, which are often bespoke and dependent on the data, model and biological question being asked, are great candidates for community-based development. We designed SeqExplainer to also serve as a platform for crowd-sourcing these types of motif analyses. We address these points in the new **Model interpretation with SeqExplainer** section of the manuscript:

"Attribution analysis, though very useful, stops short of quantifying the effect of whole motifs on model predictions. To get at the quantitative effects of such patterns, EUGENE offers a wide range of functionality for conducting *in silico* experiments with motifs of interest^{35,38}, also known as global importance analyses or GIAs¹⁴. Since the space of possible GIAs is essentially infinite, and the type of GIA used is often dependent on the data, model and biological question being asked, we provide the building blocks for GIAs in SeqExplainer, including functionality for generating background sequences and introducing perturbations (e.g., mutations, motif embedding, motif occlusion, etc.) to those sequences (**Figure 2k**). EUGENE currently offers high level functions for streamlining positional importance analysis (**Figure 2l**) and distance-dependent motif cooperativity analysis, and we anticipate adding many more common GIAs to EUGENE in the future." (page 9, paragraph 1)

3. **Memory limitations.** A major concern is that the entire dataset has to be loaded into memory. For many deep learning applications, this is not feasible even for DGX systems. PyTorch supports reading chunks of data during training and this should be easily included in EUGENE. Otherwise, this would be a major shortcoming of the package and this should be made more clear upfront within the scope of the paper (i.e. abstract and introduction).

We would first like to clarify that the whole dataset is not loaded onto memory on the GPU. GPU training is done with mini-batches and there is no requirement to store the whole dataset in GPU RAM. We have now also reimplemented EUGENE's native data structure

to support completely out-of-core training on datasets that are larger than CPU RAM (see response to Reviewer #1 comment 1).

4. Base-resolution predictive modeling. It is unclear what tools are available for working with genome-wide regression models other than reading in BigWig files. Such models, like Bassenji, BPNNet, Enformer, among others, are becoming increasingly popular. Thus, providing access to these base (or binned) resolution regression analysis would be a major contribution. While the other analyses (Figures 3-5) are quite basic (with pytorchlightning doing most of the heavy lifting), they do still comprise the most popular use cases of deep learning in this space.

As mentioned in our response to comment 1, our tool does allow for training at base-pair or binned resolution. Therefore training models on genome-wide tasks similar to those used by BPNNet, Bassenji or Enformer could be accomplished with EUGENE. We showcase this functionality in a new example use case on EUGENE's "use cases" GitHub repository (https://github.com/ML4GLand/use_cases/tree/main/BPNNet).

5. Hyperparameter search. Another major limitation is the lack of support for hyperparameter optimization. Models that are optimized for one dataset typically benefit from fine-tuning when analyzing another, smaller dataset. What functionality is there in EUGENE to facilitate this process? Weights&Biases could aid this process.

We now offer built-in support for hyperparameter search and optimization in EUGENE. We have subsequently developed a tutorial for hyperparameter optimization with EUGENE using [RayTune](https://github.com/ML4GLand/tutorials/blob/main/eugene/train/hyperparameter_optimization.ipynb) (https://github.com/ML4GLand/tutorials/blob/main/eugene/train/hyperparameter_optimization.ipynb) and cover this in the **Model instantiation, initialization and training with PyTorch and PyTorch Lightning** section of the manuscript:

"EUGENE can then be used to fit initialized architectures to datasets (with the option to perform hyperparameter optimization through the RayTune package⁶⁰) and to assess performance and generalizability on held-out test data (Figure 2g)." (page 7, paragraph 2)

It is also possible to take a model from an external source (as long as it is a PyTorch model) and perform hyperparameter optimization during fine-tuning with this functionality. We specify the details of the fine-tuning process in the response to the reviewer comment below.

6. Fine-tuning pre-trained models. There was no mention of how a pretrained model can be fine-tuned using EUGENE. Could the weights of the trunk be fixed? Could the new output layer and the parameters in the trunk be trained with different learning rates?

Fine-tuning is also possible with EUGENE and is addressed in **Supplementary Figure 2c** (copied below) and described in the **Model instantiation, initialization and training with PyTorch and PyTorch Lightning** section:

“Using LightningModules in this manner requires only minor code modifications to allow for the versatile reuse of the same architectures in different training schemes and tasks (**Supplementary Figure 2b**) and for the fine-tuning of pretrained models (**Supplementary Figure 2c**).” (page 8, paragraph 1)

Users can instantiate a new architecture in EUGENE that includes the pretrained model in the `__init__` method. You can peel layers off layers of the pretrained model to get at the feature extractors using something like:

```
layers = list(pretrained.children())[:-1].
```

The weights of the pretrained trunk can then be fixed by using the `torch.no_grad` context manager, and the other layers fine-tuned with the regular EUGENE fit or hyperoptimization functions.

Using multiple learning rates is a little more involved and would require a user to have more knowledge of PyTorch Lightning, but also can be done. One would need to create a new LightningModule to include multiple optimizers and then modify the `train_step` function to update the different weights with their respective optimizers. We will continue developing our fine-tuning library to include this in future releases.

Supplementary Figure 2. Implementing custom architectures and training tasks in

EUGENE. **a**, Creating custom architectures that are compatible with EUGENE's basic training protocol involves first inheriting from the `torch.nn.Module` class, then defining the model's layer composition (`__init__`) and the forward propagation (`forward`) method. **b**, Architectures can also be wrapped in LightningModules to allow for training protocols other than EUGENE's current built-ins. For instance, a variational autoencoder, or VAE, requires creating two functions for calculating different parts of the loss and implementing how the functions are integrated into the training function (in its most basic form). We have omitted the changes needed to define an encoder and decoder structure and how that is handled in forward. A generative adversarial network (GAN), as another example, requires implementing a multipart loss function and configuring multiple optimizers to handle the training of the generator and discriminator. **c**, Transfer learning from pretrained models is also possible in EUGENE, and can be accomplished with simple changes to an architecture's initialization and forward functions. Namely, a pretrained PyTorch model needs to be loaded in the `__init__` method and then utilized in the forward method.

7. Filter interpretability can be meaningful or not. Thus a simple cursory visualization and a TomTom search may lead to erroneous interpretations of important motifs. It is well known that Tomtom is quite inaccurate when performing a motif search with convolutional filters (Ullah & Ben-Hur, NAR, 2021, among many others). Some thoughtful guidance should be provided to promote robust science and prevent over-interpretation.

We thank the reviewer for raising this important point. We agree that learned model filters should be interpreted with care and that the caveats associated with their analyses should be made clear to users and addressed in the manuscript. We have added text in the new **Model interpretation with SeqExplainer** section of the manuscript which addresses the limitations of interpreting learned filters and cites the papers that have addressed this²⁴⁻²⁶.

"The interpretation of learned convolutional filters, commonly employed for model architectures that begin with a convolutional layer, involves using the set of sequences that significantly activate a given filter (maximally activating subsequences) to generate a position frequency matrix (PFM) (**Figure 2i**). The PFM can then be converted to a position weight matrix (PWM), visualized as a sequence logo, and annotated with tools like TomTom⁶⁸ using databases of known motifs such as JASPAR⁶⁹ or HOCOMOCO⁷⁰. Filter interpretation in this manner does have limitations. TomTom can be inaccurate when annotating motifs from learned filters^{26,71} and this analysis does not specify the importance of each filter for model predictions⁷¹. Despite these limitations, filter interpretation can be useful for hypothesis generation and for further exploration of how architecture affects learned representations⁷¹⁻⁷³." (page 8, paragraph 3)

Furthermore, we have added a section to the documentation (<https://eugene-tools.readthedocs.io/en/latest/usage-principles.html#interpret-the-filters-in-your-first-convolutional-layer>) and a tutorial dedicated to filter interpretation (https://github.com/ML4GLand/tutorials/blob/main/seqexplainer/filter_interpretation.ipynb) that describes these limitations, and cautions users against over-interpreting positive or negative results they get from this analysis.

8. Limitations of attribution analysis. Several of the attribution maps shown in this paper also seem quite noisy. This is well known that attribution maps are sensitive to local function properties and thus they may or may not be reliable. (Han et al. "Which explanation should i choose? a function approximation perspective to characterizing post hoc explanations." arXiv (2022).) Again, a thoughtful explanation could help users understand that while EUGENE provides functionality for attribution analysis, this may or may not constitute a biologically meaningful interpretation.

We thank the reviewer for pointing out another important aspect of interpreting these sequence predictors and are in complete agreement that attributions should be interpreted with care. We have added the following text to the **Model interpretation with SeqExplainer** section of the manuscript that gives a brief explanation of the limitations of attribution analysis^{31,32}:

"Attribution analysis involves using the trained model to score every nucleotide of the input on how it influences the downstream prediction for that sequence (**Figure 2j**). In SeqExplainer and EUGENE, we currently implement several common attribution approaches. These include standard *in silico* saturation mutagenesis (ISM), InputXGradient⁷⁴, DeepLIFT⁷⁴ and GradientSHAP⁷⁵, with the latter three using functionality from the Captum package⁷⁶. Attributions can also be used to validate that the model has learned representations that resemble motifs. Unlike the filter interpretability approach described above, attributions *are* directly linked to model predictions, and can naturally be extended to model the effects of single nucleotide polymorphisms. However, attributions represent a "local", often noisy^{77,78} interpretation of a single sequence, and can require clustering into "global" attributions for cleaner interpretation. In SeqExplainer, we offer wrappers for running the popular TF-ModISco algorithm⁷⁹ to accomplish this." (page 8, paragraph 4)

Furthermore, we have added a section to the documentation (<https://eugene-tools.readthedocs.io/en/latest/usage-principles.html#calculate-per-nucleotide-attributions>) and a tutorial dedicated to attribution analysis (https://github.com/ML4GLand/tutorials/blob/main/seqexplainer/attribution_analysis.ipynb) that describes these limitations, and warn users not to overinterpret positive or negative results in these attributions.

9. Analysis of prediction evolution. This analysis is quite straightforward to evolve sequences 1 mutation at a time to greedily increase predictions. But it's not clear what insights could be gained from that analysis. There should be a more thoughtful discussion on the purpose of this analysis and how to study the properties of the mutated sequences to facilitate "interpretation" of the key factors that increase predictions. I appreciate this is a software tool and not biology paper, but demonstrating functionality should include why one would do it and what information one could gain from such an analysis.

As noted in use cases sections of the manuscript, the evolution analysis can be used as another means of identifying the salient features a model has learned. We have noted that

for many of the models we have trained with EUGENE, this greedy evolution typically results in the generation of full motifs. This analysis also represents one of the simplest forms of generating novel sequences with a desired property (namely predicted activity) *in silico*³⁷. We believe that implementing this functionality in an end-to-end tool like EUGENE enables this analysis to be investigated further. We have added the following text to the **Model interpretation with SeqExplainer** section of the manuscript that highlights the aforementioned points:

"The last class of interpretability methods currently offered in EUGENE uses trained models to guide sequence evolution. We implement the simplest form of this approach that iteratively evolves a sequence by greedily inserting the mutation with the largest predicted impact at each iteration. Starting with an initial sequence (e.g. random, shuffled, etc.), this strategy can be used to evolve synthetic functional sequences⁶³ (**Figure 2m**). This style of analysis is a promising direction for further research, and can also be used for validating that the model has learned representations that resemble motifs." (page 9, paragraph 2)

We also added a section in the documentation (<https://eugene-tools.readthedocs.io/en/latest/usage-principles.html#guided-sequence-generation>) and a tutorial (https://github.com/ML4GLand/tutorials/blob/main/seqexplainer/sequence_evolution.ipynb) dedicated to describing that this analysis is still in its experimental stage.

10. The package is mainly suitable for complete beginners as more advanced users would be able to directly use the libraries (e.g. PytorchLightning, captum, logomaker) without needing the extra layer of wrapper functions. It should be made clear that the package is intended for novice users. If the desire is to engage existing researchers already in this space, then the functionality needs to incorporate more modern architectures and expanded model interpretability with *in silico* experiments.

We designed EUGENE to be most useful for data scientists who either have little experience with deep learning or have little experience with biological data, and believe the tool can help better connect these two groups.

Advanced users may nonetheless find many aspects of EUGENE useful even if they prefer to build their workflows from the existing tools in this space. EUGENE was built in a modular manner, and can therefore be seen as another tool in the toolkit, much like captum or logomaker, where a user can take advantage of the parts they want and ignore the parts they don't. For instance, a machine learning scientist might be familiar with captum and its usage, but have never worked with a BigWig file before. Such a user could use EUGENE (or just SeqData on its own) to load and preprocess their dataset, and then decide to train, evaluate and interpret the model with other packages or their own bespoke code. EUGENE can also help advanced users reduce repetitive tasks (e.g. boilerplate architecture definitions, training loops, attribution analysis, etc.) and mitigate errors in bespoke implementations. Finally, we believe that the addition of multiple new

architectures (including BPNet) and the SeqExplainer subpackage make EUGENE more attractive for advanced users.

Importantly, we also believe that advanced users can help us further develop EUGENE's functionality by adding more complex workflows and tools, making them more broadly accessible. We have added the following section to the **Discussion** mentioning this:

"We see substantial value in continuing to extend EUGENE into spaces such as these, and have designed the toolkit to allow for easy integration of this type of functionality. To continue to make bespoke methods and workflows accessible, we intend to encourage community development of EUGENE through tutorials, workshops and a dedicated user-group." (page 19, paragraph 1)

11. I think it's worth thinking about what is the real limitation of wide-spread adoption of deep learning in genomics? Is it the difficulty of stringing together python packages (pytorch lightning, captum, logomaker)? Or is it a lack of understanding on how the models should be built and how to interpret the results from the filter analysis and attribution analysis?

The inability to perform the former makes the latter inaccessible. The core goal of EUGENE is to remove barriers and bottlenecks that arise when working with these packages to enable users to learn how to appropriately architect and interpret models. Reading the latest deep learning for genomics papers only goes so far, and building the intuition for interpreting results comes from hands-on application. EUGENE is thus a tool meant for increasing the population of scientists who understand how to build and interpret deep learning models in genomics.

Minor points

1. SOTA models are not up to date - DeepBind and DeepSEA are not SOTA models.

We agree that DeepBind and DeepSEA are no longer SOTA models. We have updated the terminology we use to refer to them in the manuscript, code base and documentation. We also have added several other model architectures during review, including DanQ³⁸, Basset³⁹, ResidualBind²², and DeepSTARR¹³, Satori²⁴, scBasset⁴⁰ (<https://github.com/ML4GLand/EUGENE/tree/main/eugene/models/zoo>). The full list of available architectures can be found in the new **Supplementary Table 3**:

Architecture Name	Class	Model Class	Custom Layers	Publication	Publication Date
TutorialCNN	Basic	CNN		N/A	N/A
FCN	Basic	FCN		N/A	N/A
CNN	Basic	CNN		N/A	N/A
RNN	Basic	RNN		N/A	N/A
Hybrid	Basic	Hybrid		N/A	N/A

dsFCN	Basic	FCN	RevComp	N/A	N/A
dsCNN	Basic	CNN	RevComp	N/A	N/A
dsRNN	Basic	RNN	RevComp	N/A	N/A
dsHybrid	Basic	Hybrid	RevComp	N/A	N/A
Inception	Basic	CNN	Inception1D	N/A	N/A
DeepBind	TF binding predictor	CNN		https://www.nature.com/articles/nbt.3300	2015
ResidualBind	TF binding predictor	CNN	Residual	https://journals.plos.org/ploscompbiol/article?id=10.1371/journal.pcbi.1008925	2021
Kopp21CNN	TF binding predictor	CNN	RevComp	https://www.nature.com/articles/s41467-020-17155-y	2021
DeepSTARR	CRE activity predictor	CNN		https://www.nature.com/articles/s41588-022-01048-5	2022
Jores21CNN	CRE activity predictor	CNN	BiConv1D	https://www.nature.com/articles/s41467-021-00932-y	2021
DeepSEA	Regulatory classifier	CNN		https://www.nature.com/articles/nmeth.3547	2015
Basset	Regulatory classifier	CNN		https://genome.cshlp.org/content/26/7/990	2019
FactorizedBasset	Regulatory classifier	CNN		https://academic.oup.com/bioinformatics/article/35/14/i108/5529138	2019
DanQ	Regulatory classifier	Hybrid		https://academic.oup.com/nar/article/44/11/e107/2468300	2017
Satori	Regulatory classifier	CNN+ attention	MultiHeadAttention	https://academic.oup.com/nar/article/49/13/e77/6266414	2021
BPNet	Profile predictor	CNN		https://www.nature.com/articles/s41588-021-00782-6	2021
DeepMEL	Single cell predictor	CNN		https://genome.cshlp.org/content/30/12/1815	2020
scBasset	Single cell predictor	CNN		https://www.nature.com/articles/s41588-022-01562-8	2022

Supplementary Table 3 is also called out in the manuscript:

"We have also constructed several published architectures that represent specific configurations of these basic architectures, and made them accessible to users through single function calls (**Supplementary Table 3**)."
(page 6, paragraph 1)

2. what is flexibility of data processing? Is it straightforward to build datasets with other customizations? Eg. using DNase-seq peaks as negative examples, which is typically done. Can the models easily incorporate additional features, such as epigenetic tracks, eg. DNase-seq, as is done for several winning models in the ENCODE-DREAM challenge?

We designed EUGENE for flexibility in data processing. It supports many popular ways of loading and processing data. For the specific case mentioned, one could use a DNase peak bed file as an input to the `read_genome_fasta` function in EUGENE to extract these sequences and then assign them all a label of 0 during training. We address data processing flexibility more generally in the new **Data extraction, transformation and loading with SeqData** section of the manuscript:

"The EUGENE workflow begins with extracting data from on-disk formats. Though standardized file formats exist in regulatory genomics, their complexity can make creating model-ready datasets non-trivial. To address this in EUGENE, we created the standalone subpackage `SeqData` (<https://github.com/ML4GLand/SeqData>) which flexibly and efficiently reads data from a variety of file formats, including CSV/TSV (tabular), FASTA, BED, BAM, and BigWig (**Figure 2a**, top). The versatility of `SeqData` enables the generation of many custom datasets from combinations of these file types, including several commonly utilized in regulatory genomics. These include datasets derived from combinations of tabular and FASTA files that are suitable for single and multi-task regression and classification (e.g. DeepSTARR³⁵), datasets from genomic coordinates defined in BED files suitable for multi-task binary classification (e.g. DeepSEA¹⁵ or Sei²³), and datasets from multiple BigWigs and BED files suitable for binned or base-pair resolution regression (e.g. Basenji¹⁸ and BPNNet³⁸ respectively). EUGENE also supplies a growing collection of hand-curated datasets available via the `SeqDatasets` subpackage (<https://github.com/ML4GLand/SeqDatasets>) (**Supplementary Table 1**) that can be downloaded and subsequently loaded into a workflow via a single function call (**Figure 2a**, bottom)." (pages 3 and 5)

To address the final question, we designed EUGENE primarily to be used for modeling sequence as the sole input, so built-in models will not be able to handle the incorporation of epigenetic tracks as features. However, `SeqData` is flexible enough to provide both tracks and sequences as input. If a user designs a custom architecture to take in both sequence and epigenetic tracks as input, this model can be trained, evaluated and interpreted with EUGENE.

3. The order of the basic conv module is conv, activation, pool (optional), dropout (optional), batch norm (optional). From model to model, the order of these can be employed differently, eg. Basset, DeepSTARR.

We thank the reviewer for pointing this out. We have updated our codebase with an "order" argument to allow for different orderings of the layers in a convolutional block (https://github.com/ML4GLand/EUGENE/blob/main/eugene/models/base_blocks.py#L29C51-L29C51). A user can therefore instantiate a DeepSEA conv block with "conv-act-pool-dropout", a DeepSTARR conv block with "conv-norm-act-pool", or any other combination of conv, activation, pool, norm and drop out. This is now mentioned in the **Model instantiation, initialization and training with PyTorch and PyTorch Lightning** section of the manuscript:

"Designing and training neural networks for regulatory genomics requires a comprehensive library of architecture building blocks. EUGENE builds on the PyTorch library of neural network layers by adding several useful layers such as inception and residual layers. Additionally, EUGENE provides flexible functions for instantiating common "blocks" and "towers" that are composed of heterogeneous sets of layers arranged in a predefined or adaptable order. For instance, a convolutional block (Conv1DBlock in EUGENE) often comprises convolutional, normalization, activation, and dropout layers in different orderings depending on the model and task (**Figure 2e**, top)." (pages 5 and 7)

4. Should cite captum as sources for attribution maps.

We thank the reviewer for noting this and have added a citation for captum when it is mentioned in the **Model interpretation with SeqExplainer** section of the manuscript:

"These include standard *in silico* saturation mutagenesis (ISM), InputXGradient⁷⁴, DeepLIFT⁷⁴ and GradientSHAP⁷⁵, with the latter three using functionality from the Captum package⁷⁶" (page 8, paragraph 4)

Remarks on code availability

The code is organized well. I did not try to install or run the code.

Reviewer #3

Remarks to the Author

In this paper, Kile et al. presented a python package, EUGENE, for training genomic sequence deep learning models. The package is based on PyTorch, a popular deep learning framework. Compared to existing toolkits for sequence-based deep learning software, EUGENE is designed to be simple and extensible. The authors then applied the package to three prediction tasks published previously.

Although I appreciate the non-trivial software development efforts from the author, I have concerns regarding the novelty of this work, and how it positions itself for novice and expert ML users. The use cases are over-simplified. My comments are below.

Major points

1. The EUGENE package presents two python class objects SeqData and BaseModel. This requires considerable knowledge of Python object-oriented programming, which is relatively less user-friendly than text configuration-based framework, such as Selene. Therefore, it poses a barrier to novice ML users.

We designed EUGENE primarily for those data scientists who are just getting into deep learning for genomics that either have little experience with deep learning or have little experience with handling genomics data. Though text configuration-based frameworks require less knowledge of programming, it is becoming safer each day to assume that most such researchers are comfortable working with Python in Jupyter notebooks. In addition, encouraging use of a Python API empowers users to utilize integrative development environments (IDEs) which can show them function signatures, type hints, documentation strings, as well as autocomplete parameters and arguments, check misspellings, and perform static type checking (e.g. via MyPy, Pylance). In effect, using the Python API is akin to filling out a form that is interactive, autocorrecting, and self-explanatory. These features are available out-of-the-box from mainstream IDEs (e.g. PyCharm, VSCode), but are less accessible or otherwise not available with a text (e.g. JSON, YAML) configuration-based framework.

Configuration files can be useful (EUGENE uses them as a way to define model architectures), but they have their limitations. They can be challenging to read if they contain many parameters and difficult to understand if each parameter is individually complex. While they often make sense for more robust and standardized workflows (e.g. processing sequencing reads), they are less desirable for running deep learning experiments, as they do not offer as much interaction with each stage of the workflow. Furthermore, although Selene does use a text configuration-based framework, their configuration files often contain references to object-oriented programming. Take the following example from their documentation

(<https://selene.flatironinstitute.org/master/overview/cli.html#train>) which is essentially a call to a model in Python:

```
train_model: !obj:selene_sdk.TrainModel {
  batch_size: 64,
  max_steps: 960000,
  report_stats_every_n_steps: 32000,
  save_checkpoint_every_n_steps: 1000,
  save_new_checkpoints_after_n_steps: 640000,
  n_validation_samples: 64000,
  n_test_samples: 960000,
  cpu_n_threads: 32,
  use_cuda: True,
  data_parallel: True,
  logging_verbosity: 2,
  metrics: {
    roc_auc: !import sklearn.metrics.roc_auc_score,
    average_precision: !import sklearn.metrics.average_precision_score
  },
  checkpoint_resume: False
}
```

2. For users who are expert in python and/or ML, what's the advantage of using EUGENE's 'BaseModel' class compared to pytorch-lightning's 'LightningModule'?

We thank the reviewer for pointing out this potential point of confusion. We have now refactored BaseModel into a set of standard LightningModules. Each of these LightningModules is meant to 1) define the types of architectures it can train, 2) standardize the way a user interacts with those architectures in EUGENE, and 3) reduce boilerplate PyTorch and PyTorch Lightning code. For example, we implemented SequenceModule (the replacement for BaseModel) to expect an architecture that ingests a single tensor (usually one-hot encoded DNA sequences) as input and outputs a single tensor. The SequenceModule defines how this class of models should be trained (including the loss function and optimizer), what metrics should be reported, and how inference should be handled. As a result, any PyTorch model that follows this contract can be trained using SequenceModule. We have also implemented a ProfileModule that handles BPNets style training, where the model has multiple output tensors ("heads"), can take in optional control inputs, and uses multiple loss functions. We now describe the use of LightningModules in more detail in the new **Model instantiation, initialization and training with PyTorch and PyTorch Lightning** section of the manuscript:

"For training, EUGENE uses the PyTorch Lightning (PL) framework⁸¹ and programmatic objects called LightningModules. Each EUGENE LightningModule delineates the architecture types it can train and standardizes boilerplate tasks for those architectures (e.g optimizer configuration, metric logging, etc.). For instance, the primary LightningModule in EUGENE, termed SequenceModule (**Figure 2h**), anticipates training an architecture that takes in a single tensor (typically one-hot encoded DNA sequences)

and delivers a single tensor output. We have also implemented a ProfileModule for BPNet-style³⁸ training, where models produce multiple tensor outputs (or "heads"), accept optional control inputs, and use multiple loss functions (https://github.com/ML4GLand/use_cases/blob/main/BPNet/train_eugene.ipynb). Using LightningModules in this manner requires only minor code modifications to allow for the reuse of the same architectures in different training schemes and tasks (**Supplementary Figure 2b**) and for the fine-tuning of pretrained models (**Supplementary Figure 2c**). We plan on continuing to develop our library of LightningModules for different training schemes, including adversarial learning⁶², generative modeling⁶³, language modeling⁶⁴, and more." (pages 7 and 8)

3. The functionality and API of EUGENE is highly similar to Janggu. Although Janggu is based on Keras, while EUGENE is based on PyTorch, this backend difference does not seem scientifically significant to me.

EUGENE draws inspiration from predecessor packages such as Janngu, yet differs in several ways:

- The use of PyTorch and PyTorch Lightning over Keras as the backend should not be overlooked. PyTorch and PyTorch Lightning are among the fastest developing frameworks within the deep learning research community (see https://mlcontests.com/state-of-competitive-machine-learning-2022/?utm_source=substack&utm_medium=email#winning-toolkit under "No competition for PyTorch subheading"). We provide access to many data types, models and methods to PyTorch users in genomics, and can continue to take advantage of functionality added to these frameworks as they grow.
- EUGENE offers more flexibility in data loading than Janggu once a read function has been called. Janggu requires decisions to be made about a dataset up front prior to loading (e.g. order of encoding, binning resolution, etc.). SeqData is more flexible thanks to lazy computations provided by Dask arrays. A user can load the rawest form of the data and manipulate it in several different ways without having to reload or load the dataset into memory (see response to Reviewer #1 comment 1)
- EUGENE offers a larger interpretability suite than Janngu, which is limited to a single, somewhat outdated feature attribution method (integrated gradients)
- Janggu offers little in the way of built-in models or support for architecture design, while EUGENE has an entire library dedicated to this
- EUGENE is more extendable than Janggu, which appears to be no longer maintained having no updates since September 2021.
- EUGENE's core data structure relies on widely used and NumFOCUS supported packages. EUGENE stands to benefit from the continued development of those packages and their roadmaps which include: (1) support for out-of-core sparse n-dimensional arrays (<https://github.com/zarr-developers/zarr-specs/issues/245>), which would open the door to seamlessly and efficiently working with single-cell and Hi-C data. (2) Flexible indexing

<https://docs.xarray.dev/en/stable/roadmap.html#flexible-indexes>) which would enable support for indexing via interval trees and fast range queries (i.e. retrieving all sequences within a range).

4. The input sequence length in the case studies are 170bp, 200bp, 200bp, for each of the three use cases, respectively. They are substantially smaller than current powerful models, such as Enformer (200kb), Sei (4096bp), or even earlier works such as Basenji (131kb) and DeepSEA (1000bp). The authors should consider tasks with enlarged input sequence size to match the current standards.

These tasks are all possible within the EUGENE framework. We show data loading and training times for 10kb long sequences in **Supplementary Figure 1** (copied below). We also have created many code examples for using EUGENE on longer sequence examples that can be found at the accompanying "use cases" repository (https://github.com/ML4GLand/use_cases)

Supplementary Figure 1. Peak memory usage and batch processing time for datasets with increasing numbers of 10,000bp sequences. a, Peak memory usage against number of sequences for datasets loaded into memory versus datasets loaded “out-of-core.” **b,** Median time in seconds taken for processing a batch of 100 sequences using the same datasets as in **a.** Error bars indicate interquartile ranges across up to 100 random batches. Both axes in **a** and **b** are on the log₁₀ scale. All analyses were performed using a chunk size of 4096 along the sequence dimension on a machine with 2 CPU cores and 64GB of RAM.

- To show real-world application, the authors should also consider more complex architectures that have been shown effective in genomics recently, such as residual connections, inception layers, and attention layers.

We thank the reviewer for this suggestion. We have now added residual layers, inception layers and self-attention layers to EUGENE (https://github.com/ML4GLand/EUGENE/blob/main/eugene/models/base_layers.py#L174) as well as several more complex full architectures. This is reflected in **Supplementary Table 3** (shown below) and in the **Model instantiation, initialization and training with PyTorch and PyTorch Lightning** section of the manuscript:

Supplementary Table 3. Built-in architectures in EUGENE's models module

Architecture Name	Class	Model Class	Custom Layers	Publication	Publication Date
TutorialCNN	Basic	CNN		N/A	N/A
FCN	Basic	FCN		N/A	N/A
CNN	Basic	CNN		N/A	N/A
RNN	Basic	RNN		N/A	N/A
Hybrid	Basic	Hybrid		N/A	N/A
dsFCN	Basic	FCN	RevComp	N/A	N/A
dsCNN	Basic	CNN	RevComp	N/A	N/A
dsRNN	Basic	RNN	RevComp	N/A	N/A
dsHybrid	Basic	Hybrid	RevComp	N/A	N/A
Inception	Basic	CNN	Inception1D	N/A	N/A
DeepBind	TF binding predictor	CNN		https://www.nature.com/articles/nbt.3300	2015
ResidualBind	TF binding predictor	CNN	Residual	https://journals.plos.org/ploscompbiol/article?id=10.1371/journal.pcbi.1008925	2021
Kopp21CNN	TF binding predictor	CNN	RevComp	https://www.nature.com/articles/s41467-020-17155-y	2021
DeepSTARR	CRE activity predictor	CNN		https://www.nature.com/articles/s41588-022-01048-5	2022
Jores21CNN	CRE activity predictor	CNN	BiConv1D	https://www.nature.com/articles/s41467-021-00932-y	2021
DeepSEA	Regulatory classifier	CNN		https://www.nature.com/articles/nmeth.3547	2015
Basset	Regulatory classifier	CNN		https://genome.cshlp.org/content/26/7/990	2019
FactorizedBasset	Regulatory classifier	CNN		https://academic.oup.com/bioinformatics/article/35/14/i108/5529138	2019
DanQ	Regulatory classifier	Hybrid		https://academic.oup.com/nar/article/44/11/e107/2468300	2017
Satori	Regulatory classifier	CNN+attention	MultiHeadAttention	https://academic.oup.com/nar/article/49/13/e77/6266414	2021
BPNet	Profile predictor	CNN		https://www.nature.com/articles/s41588-021-00782-6	2021

DeepMEL	Single cell predictor	CNN	https://genome.cshlp.org/content/30/12/1815	2020
scBasset	Single cell predictor	CNN	https://www.nature.com/articles/s41592-022-01562-8	2022

“Designing and training neural networks for regulatory genomics requires a comprehensive library of architecture building blocks. EUGENE builds on the PyTorch library of neural network layers by adding several useful layers such as inception and residual layers.” (page 5, paragraph 4)

“We have also constructed several published architectures that represent specific configurations of these basic architectures, and made them accessible to users through single function calls (**Supplementary Table 3**).” (page 7, paragraph 1)

We also feel that this is an ideal area for community-based development, as bespoke layers are common in deep learning and can be dependent on the type of training tasks and data one is working with. We also want to note that relatively simple convolutional models are often sufficient for even complex sequence-based prediction tasks⁴⁴.

- As a high-level software package, it is also important to show if a model built by EUGENE can train as efficiently as in PyTorch-lightning and in native PyTorch, such as GPU utilization rate, GPU time.

EUGENE makes direct calls to PyTorch Lightning code, so we do not anticipate any performance differences from PyTorch Lightning. To test this, we benchmarked the GPU efficiency of EUGENE against PyTorch Lightning training and a comparable native PyTorch training loop (no validation set processing) on the JunD binding use case presented in the manuscript. We trained the Kopp21CNN model described in the manuscript with native PyTorch, PyTorch Lightning with and without a validation set, and the default EUGENE “fit” method. We monitored GPU utilization over walltime using Weights and Biases (WandB) and included the output from WandB below.

We can see that GPU utilization is comparable for all runs and that training time is comparable for EUGENE and PyTorch Lightning. Native PyTorch is indeed faster than PyTorch Lightning (with and without a validation loop) and EUGENE, but this is a known constant overhead that comes with additional features of PyTorch Lightning (e.g. logging, pre-training checks etc.). We include GitHub issue links that document this below. These speed costs are often negligible for training larger models on larger datasets.

- Speed comparisons to native PyTorch:
 - <https://github.com/Lightning-AI/lightning/issues/8857>
 - <https://github.com/Lightning-AI/lightning/issues/6196>
- Start-up cost for PyTorch Lightning:
 - <https://github.com/Lightning-AI/lightning/issues/9304>

References

1. Kopp, W., Monti, R., Tamburrini, A., Ohler, U. & Akalin, A. Deep learning for genomics using Janggu. *Nat. Commun.* **11**, 3488 (2020).
2. Hoyer, S. & Hamman, J. xarray: N-D labeled Arrays and Datasets in Python. **5**, 10 (2017).
3. Team, D. D. Dask: Library for dynamic task scheduling. Preprint at <https://dask.org> (2016).
4. Miles, A. *et al.* *zarr-developers/zarr-python: v2.15.0.* (2023). doi:10.5281/zenodo.8039103.
5. Baker, E. A. G. *et al.* emObject: domain specific data abstraction for spatial omics. *bioRxiv* 2023.06.07.543950 (2023) doi:10.1101/2023.06.07.543950.
6. Marconato, L. *et al.* SpatialData: an open and universal data framework for spatial omics. *bioRxiv* 2023.05.05.539647 (2023) doi:10.1101/2023.05.05.539647.
7. Liu, H. *et al.* DNA methylation atlas of the mouse brain at single-cell resolution. *Nature* **598**, 120–128 (2021).
8. Avsec, Ž. *et al.* Effective gene expression prediction from sequence by integrating long-range interactions. *Nat. Methods* **18**, 1196–1203 (2021).
9. Kelley, D. R. *et al.* Sequential regulatory activity prediction across chromosomes with convolutional neural networks. *Genome Res* **28**, 739–750 (2018).
10. Teufel, F. *et al.* GraphPart: Homology partitioning for biological sequence analysis. *bioRxiv* 2023.04.14.536886 (2023) doi:10.1101/2023.04.14.536886.
11. Whalen, S., Schreiber, J., Noble, W. S. & Pollard, K. S. Navigating the pitfalls of applying machine learning in genomics. *Nat. Rev. Genet.* (2021) doi:10.1038/s41576-021-00434-9.
12. Urban, G., Torrisi, M., Magnan, C. N., Pollastri, G. & Baldi, P. Protein profiles: Biases and protocols. *Comput. Struct. Biotechnol. J.* **18**, 2281–2289 (2020).
13. de Almeida, B. P., Reiter, F., Pagani, M. & Stark, A. DeepSTARR predicts enhancer activity from DNA sequence and enables the de novo design of synthetic enhancers. *Nat. Genet.* **54**, 613–624 (2022).

14. Zhou, J. & Troyanskaya, O. G. Predicting effects of noncoding variants with deep learning-based sequence model. *Nat. Methods* **12**, 931–934 (2015).
15. Avsec, Ž. *et al.* Base-resolution models of transcription-factor binding reveal soft motif syntax. *Nat. Genet.* **53**, 354–366 (2021).
16. Barker, M. *et al.* Introducing the FAIR Principles for research software. *Sci Data* **9**, 622 (2022).
17. Wilkinson, M. D. *et al.* The FAIR Guiding Principles for scientific data management and stewardship. *Sci Data* **3**, 160018 (2016).
18. Koo, P. K. & Ploenzke, M. Deep learning for inferring transcription factor binding sites. *Curr Opin Syst Biol* **19**, 16–23 (2020).
19. Novakovsky, G., Dexter, N., Libbrecht, M. W., Wasserman, W. W. & Mostafavi, S. Obtaining genetics insights from deep learning via explainable artificial intelligence. *Nat. Rev. Genet.* 1–13 (2022) doi:10.1038/s41576-022-00532-2.
20. Talukder, A., Barham, C., Li, X. & Hu, H. Interpretation of deep learning in genomics and epigenomics. *Brief. Bioinform.* **22**, (2021).
21. Lee, N. K., Tang, Z., Toneyan, S. & Koo, P. K. EvoAug: improving generalization and interpretability of genomic deep neural networks with evolution-inspired data augmentations. *Genome Biol.* **24**, 105 (2023).
22. Koo, P. K., Majdandzic, A., Ploenzke, M., Anand, P. & Paul, S. B. Global importance analysis: An interpretability method to quantify importance of genomic features in deep neural networks. *PLoS Comput. Biol.* **17**, e1008925 (2021).
23. Moritz, P. *et al.* Ray: A Distributed Framework for Emerging AI Applications. *arXiv [cs.DC]* (2017).
24. Ullah, F. & Ben-Hur, A. A self-attention model for inferring cooperativity between regulatory features. *Nucleic Acids Res.* **49**, e77 (2021).
25. Koo, P. K. & Ploenzke, M. Improving representations of genomic sequence motifs in

- convolutional networks with exponential activations. *Nat Mach Intell* **3**, 258–266 (2021).
26. Koo, P. K. & Eddy, S. R. Representation learning of genomic sequence motifs with convolutional neural networks. *PLoS Comput. Biol.* **15**, e1007560 (2019).
 27. Gupta, S., Stamatoyannopoulos, J. A., Bailey, T. L. & Noble, W. S. Quantifying similarity between motifs. *Genome Biol.* **8**, R24 (2007).
 28. Castro-Mondragon, J. A. *et al.* JASPAR 2022: the 9th release of the open-access database of transcription factor binding profiles. *Nucleic Acids Res.* **50**, D165–D173 (2022).
 29. Kulakovskiy, I. V. *et al.* HOCOMOCO: towards a complete collection of transcription factor binding models for human and mouse via large-scale ChIP-Seq analysis. *Nucleic Acids Res.* **46**, D252–D259 (2018).
 30. Ploenzke, M. S. & Irizarry, R. A. Interpretable Convolution Methods for Learning Genomic Sequence Motifs. *bioRxiv* 411934 (2018) doi:10.1101/411934.
 31. Han, T., Srinivas, S. & Lakkaraju, H. Which Explanation Should I Choose? A Function Approximation Perspective to Characterizing Post Hoc Explanations. *arXiv [cs.LG]* (2022).
 32. Majdandzic, A., Rajesh, C. & Koo, P. K. Correcting gradient-based interpretations of deep neural networks for genomics. *bioRxiv* 2022.04.29.490102 (2022) doi:10.1101/2022.04.29.490102.
 33. Shrikumar, A., Greenside, P., Shcherbina, A. & Kundaje, A. Not Just a Black Box: Learning Important Features Through Propagating Activation Differences. *arXiv [cs.LG]* (2016).
 34. Lundberg, S. M. & Lee, S.-I. A Unified Approach to Interpreting Model Predictions. in *Advances in Neural Information Processing Systems 30* (eds. Guyon, I. *et al.*) 4765–4774 (Curran Associates, Inc., 2017).
 35. Kokhlikyan, N. *et al.* Captum: A unified and generic model interpretability library for PyTorch. *arXiv [cs.LG]* (2020).
 36. Shrikumar, A. *et al.* Technical Note on Transcription Factor Motif Discovery from Importance Scores (TF-MoDISco) version 0.5.6.5. *arXiv [cs.LG]* (2018).

37. Taskiran, I. I., Spanier, K. I., Christiaens, V., Mauduit, D. & Aerts, S. Cell type directed design of synthetic enhancers. *bioRxiv* 2022.07.26.501466 (2022)
doi:10.1101/2022.07.26.501466.
38. Quang, D. & Xie, X. DanQ: a hybrid convolutional and recurrent deep neural network for quantifying the function of DNA sequences. *Nucleic Acids Res.* **44**, e107 (2016).
39. Kelley, D. R., Snoek, J. & Rinn, J. L. Basset: learning the regulatory code of the accessible genome with deep convolutional neural networks. *Genome Res* **26**, 990–999 (2016).
40. Yuan, H. & Kelley, D. R. scBasset: sequence-based modeling of single-cell ATAC-seq using convolutional neural networks. *Nat. Methods* (2022) doi:10.1038/s41592-022-01562-8.
41. Falcon, W. *et al.* *PyTorchLightning/pytorch-lightning: 0.7.6 release.* (Zenodo, 2020).
doi:10.5281/ZENODO.3828935.
42. Koo, P. K., Qian, S., Kaplun, G., Volf, V. & Kalimeris, D. Robust Neural Networks are More Interpretable for Genomics. *bioRxiv* 657437 (2019) doi:10.1101/657437.
43. Ji, Y., Zhou, Z., Liu, H. & Davuluri, R. V. DNABERT: pre-trained Bidirectional Encoder Representations from Transformers model for DNA-language in genome. *Bioinformatics* (2021) doi:10.1093/bioinformatics/btab083.
44. Penzar, D. *et al.* LegNet: resetting the bar in deep learning for accurate prediction of promoter activity and variant effects from massive parallel reporter assays. *bioRxiv* 2022.12.22.521582 (2022) doi:10.1101/2022.12.22.521582.

Decision Letter, first revision:

Date: 31st August 23 18:19:58
Last Sent: 31st August 23 18:19:58
Triggered By: Fernando Chirigati
From: fernando.chirigati@us.nature.com
To: hkcarter@ucsd.edu
CC: computacionalscience@nature.com
BCC: fernando.chirigati@us.nature.com
Subject: AIP Decision on Manuscript NATCOMPUTSCI-23-0044A-Z
Message: Our ref: NATCOMPUTSCI-23-0044A-Z

31st August 2023

Dear Dr. Carter,

Thank you for submitting your revised manuscript "EUGENE: A Python toolkit for predictive analyses of regulatory sequences" (NATCOMPUTSCI-23-0044A-Z). It has now been seen by the original referees and their comments are below. The reviewers find that the paper has improved in revision, and therefore we'll be happy in principle to publish it in Nature Computational Science, pending minor revisions to satisfy the referees' final requests and to comply with our editorial and formatting guidelines.

We are now performing detailed checks on your paper and will send you a checklist detailing our editorial and formatting requirements in about a week. Please **do not upload** the final materials and make any revisions until you receive this additional information from us.

TRANSPARENT PEER REVIEW

Nature Computational Science offers a transparent peer review option for original research manuscripts. We encourage increased transparency in peer review by publishing the reviewer comments, author rebuttal letters and editorial decision letters if the authors agree. Such peer review material is made available as a supplementary peer review file. **Please remember to choose, using the manuscript system, whether or not you want to participate in transparent peer review.**

Thank you again for your interest in Nature Computational Science. Please do not hesitate to contact me if you have any questions.

Best,
Fernando

--

Fernando Chirigati, PhD
Chief Editor, Nature Computational Science
Nature Portfolio

ORCID

Reviewer #1 (Remarks to the Author):

My major comments were addressed, and it seems that many/most of the comments from the other reviewers were also addressed. I recommend acceptance. Specifically, the authors have substantially improved the manuscript. The implementation of the out-of-core loading and training in the subpackage SeqData and the improved explanations on the use cases and supported data types was appreciated, as well as the implemented train-test split by homologues. Code, documentation, and tutorials have improved substantially in the new version.

Reviewer #2 (Remarks to the Author):

The authors have done a great job of addressing most of my original concerns. Now, Eugene covers a more comprehensive set of functionality that separates it from other frameworks. Eugene is currently in a state that could be beneficial to a much wider user-base, which was one of my main concerns with the initial submission. With further development and community support, I can envision that Eugene could become widely adopted, enabling more researchers access to end-to-end data analysis of deep learning models for functional genomics. The resubmission is well written and I only have a few suggestions for the code repository -- I don't believe another round of review is needed.

Minor concerns:

There are a few issues with some of the tutorials, which may be due to not having all of the right requirements set up:

1. tutorials/eugene/models/instantiating_models.ipynb
cell [31] models.SequenceModule throws an error
cell [37] load_config not defined
cell [44] evoaug_analysis not found
cell [46] utils not defined2.

2. tutorials/eugene/end_to_end/basic_usage_tutorial.ipynb
cells [16], [32], [37] plot not printing

Suggestion 1: It would be helpful to have either a “roadmap” of sequential order that files should be run to follow the tutorials or provide a minimal set of intermediate files so that the notebooks can work independently.

Suggestion 2: Due to package requirements and potential compatibility issues, it could be beneficial to host additional tutorial notebooks on colab, where the analysis can be run without requiring installation of finicky environments or containers.

Reviewer #2 (Remarks on code availability):

Overall, the code and readthedocs is complete. Some of the notebooks are still a work in progress but as long as the authors are dedicated to their completion and maintenance, these minor issues should not hold the publication up.

Reviewer #3 (Remarks to the Author):

I'd like to thank the authors for their efforts to address my concerns and to add new functionalities to the software. I believe the presented software in its current form provides a useful tool and support its publication in Nature Computational Science.

As a computational toolkit, the evaluation of code availability is essential. I did notice some of the more complex models listed by the authors were not implemented, or with bugs (at least 4 out of the 7 examples under use_cases). These are detailed in the code availability section below.

Reviewer #3 (Remarks on code availability):

- use_cases/scBasset : not implemented
- use_cases/Basenji : not implemented
- use_cases/DeepMEL : not implemented
- use_cases/DeepSTARR : notebooks have many error messages/bugs

Author Rebuttal, first revision:

Summary of responses

We thank the reviewers for their comments and suggestions throughout the review process. We have addressed the remaining comments in the point-by-point responses below.

Reviewer #1

Remarks to the Author

My major comments were addressed, and it seems that many/most of the comments from the other reviewers were also addressed. I recommend acceptance. Specifically, the authors have substantially improved the manuscript. The implementation of the out-of-core loading and training in the subpackage SeqData and the improved explanations on the use cases and supported data types was appreciated, as well as the implemented train-test split by homologues. Code, documentation, and tutorials have improved substantially in the new version.

We thank the reviewer for the many helpful suggestions that have strengthened our manuscript.

Remarks on code availability

See above.

Reviewer #2

Remarks to the Author

The authors have done a great job of addressing most of my original concerns. Now, Eugene covers a more comprehensive set of functionality that separates it from other frameworks. Eugene is currently in a state that could be beneficial to a much wider user-base, which was one of my main concerns with the initial submission. With further development and community support, I can envision that Eugene could become widely adopted, enabling more researchers access to end-to-end data analysis of deep learning models for functional genomics. The resubmission is well written and I only have a few suggestions for the code repository -- I don't believe another round of review is needed.

Minor points

There are a few issues with some of the tutorials, which may be due to not having all of the right requirements set up:

1. tutorials/eugene/models/instantiating_models.ipynb
cell [31] models.SequenceModule throws an error

cell [37] load_config not defined

cell [44] evoaug_analysis not found

cell [46] utils not defined2.

We thank the reviewer for pointing out these errors. The first two were simple syntax errors that we have fixed. The latter two are likely due to the fact that the evoaug and evoaug_analysis packages are not EUGENE dependencies and therefore must be installed manually. We have added code for installing these packages to the tutorials that should fix these errors.

We have also added instructions to the tutorials GitHub repository README.md file (<https://github.com/ML4GLand/tutorials>) for how to open notebooks hosted in GitHub repositories on Colab. This will help avoid requirement and dependency issues in the future.

2. tutorials/eugene/end_to_end/basic_usage_tutorial.ipynb
cells [16], [32], [37] plot not printing

We thank the reviewer for pointing out this issue. We have noticed that whether or not a plot is displayed in a Jupyter notebook is somewhat specific to the user's environment. In the environments we have tested, for instance, we have not experienced issues with plot display when the `%matplotlib inline` command is run. To properly debug this problem, it may be helpful to raise a GitHub issue that includes the specific environment configuration used. Nevertheless, we will continue to look into this.

3. Suggestion 1: It would be helpful to have either a "roadmap" of sequential order that files should be run to follow the tutorials or provide a minimal set of intermediate files so that the notebooks can work independently.

We agree with the reviewer that a roadmap would be a useful addition to the tutorials repository and that all of the necessary intermediate files should be included there as well. We have updated the tutorials repository to address this as follows:

- We have updated the directories in the repository to contain the minimal set of intermediate files necessary for running notebooks independently (when possible). This includes but is not limited to:
 - "data" -- Auxiliary data needed for tutorials(e.g. MEME files)
 - "configs" -- YAML files for instantiating architectures used in the tutorials
 - "models" -- Checkpoints for model weights used in the tutorialsWe provide instructions on how to download files that are too large to be hosted on GitHub in specific notebooks.
- We have updated the repository README.md file (<https://github.com/ML4GLand/tutorials/blob/main/README.md>) to serve as a guide for running tutorials. It now includes a recommended sequential order for running the available tutorials, as well as instructions for running notebooks on Google Colab.

4. Suggestion 2: Due to package requirements and potential compatibility issues, it could be beneficial to host additional tutorial notebooks on colab, where the analysis can be run without requiring installation of finicky environments or containers.

We thank the reviewer for this suggestion. We have provided instructions in the tutorial repository README.md file for how to open notebooks hosted on GitHub via a Colab instance (<https://github.com/ML4GLand/tutorials/tree/main#running-tutorials-on-colab>). This has the added benefit that these notebooks can continue to be version controlled on GitHub, instead of needing to update those hosted on Colab as changes are made.

Remarks on code availability

Overall, the code and readthedocs is complete. Some of the notebooks are still a work in progress but as long as the authors are dedicated to their completion and maintenance, these minor issues should not hold the publication up.

Reviewer #3

Remarks to the Author

I'd like to thank the authors for their efforts to address my concerns and to add new functionalities to the software. I believe the presented software in its current form provides a useful tool and support its publication in Nature Computational Science.

As a computational toolkit, the evaluation of code availability is essential. I did notice some of the more complex models listed by the authors were not implemented, or with bugs (at least 4 out of the 7 examples under `use_cases`). These are detailed in the code availability section below.

Remarks on code availability

We thank the reviewer for pointing out the gaps in our `use_cases` repository (https://github.com/ML4GLand/use_cases). As described in more detail below, we have taken several steps to address the points brought up by the reviewer for each of the 4 examples, and will continue to improve upon these use cases as the software matures.

- `use_cases/scBasset` : not implemented

- We have added code that shows how to preprocess the dataset provided by Chen *et al.* for training models with both EUGENE and code from the `scBasset` package (<https://github.com/calico/scBasset>).
- We have added code that shows how to train a model on the Chen *et al.* dataset using the `scBasset` package.
- We have added code that shows how to train a model on the Chen *et al.* dataset using EUGENE.
- We will be adding code in the future that shows how to benchmark these against each other, and against the pretrained model deposited in the Kipoi repository (<https://kipoi.org/models/scbasset/>)

- use_cases/Basenji : not implemented

- We unfortunately do not have the resources to retrain the Basenji2 model (though it is possible to do with EUGENE with the proper resources), and instead have provided examples of how to interact with the pretrained model:
 - We have added code that shows how to use EUGENE (and other sources) for inference with a pretrained Basenji model.
 - We will soon be adding code that shows how to interpret this pretrained model with SeqExplainer.
 - We will be adding code in the future that shows how to fine-tune the pretrained model on a new dataset with EUGENE.
 - We will be adding code in the future that shows how to benchmark these models against the pretrained Basenji in the Kipoi repository (<https://kipoi.org/models/Basenji/>).

- use_cases/DeepMEL : not implemented

- We have added code that shows how to preprocess the PBMC 3k Multiome dataset provided by 10x (<https://www.10xgenomics.com/resources/datasets/pbmc-from-a-healthy-donor-granulocytes-removed-through-cell-sorting-3-k-1-standard-2-0-0>) for training models using pycisTopic and EUGENE.
- We have added code that shows how to train, evaluate and interpret a DeepMEL model on this dataset using EUGENE.

- use_cases/DeepSTARR : notebooks have many error messages/bugs

- The errors and bugs are mainly due to issues encountered when using Kipoi to interpret pretrained models deposited by others. We have removed these errors and instead have included documentation warning that these errors exist in their respective notebooks.
- We have added code that shows how to train, evaluate and interpret a DeepSTARR model with EUGENE.
- We also have code that shows how to perform a distance dependent cooperativity analysis with a model trained using the EvoAug codebase (<https://github.com/p-koo/evoaug>).

Final Decision Letter:**Date:** 27th September 23 18:10:39**Last Sent:** 27th September 23 18:10:39**Triggered By:** Fernando Chirigati**From:** fernando.chirigati@us.nature.com**To:** hkcarter@ucsd.edu**BCC:** computacionalscience@nature.com,fernando.chirigati@us.nature.com,rjsproduction@springernature.com,rjsart@springernature.com**Subject:** Decision on Nature Computational Science manuscript NATCOMPUTSCI-23-0044B**Message** Dear Professor Carter,

:

We are pleased to inform you that your Brief Communication "Predictive analyses of regulatory sequences with EUGENE" has now been accepted for publication in Nature Computational Science.

Once your manuscript is typeset, you will receive an email with a link to choose the appropriate publishing options for your paper and our Author Services team will be in touch regarding any additional information that may be required.

Please note that *Nature Computational Science* is a Transformative Journal (TJ). Authors may publish their research with us through the traditional subscription access route or make their paper immediately open access through payment of an article-processing charge (APC). Authors will not be required to make a final decision about access to their article until it has been accepted. [Find out more about Transformative Journals](https://www.springernature.com/gp/open-research/transformative-journals)

Authors may need to take specific actions to achieve [compliance with funder and institutional open access mandates](https://www.springernature.com/gp/open-research/funding/policy-compliance-faqs). If your research is supported by a funder that requires immediate open access (e.g. according to [Plan S principles](https://www.springernature.com/gp/open-research/plan-s-compliance)) then you should select the gold OA route, and we will direct you to the compliant route where possible. For authors selecting the subscription publication route, the journal's standard licensing terms will need to be accepted, including [self-archiving policies](https://www.springernature.com/gp/open-research/policies/journal-policies). Those licensing terms will supersede any other terms that the author or any third party may assert apply to any version of the manuscript.

Acceptance of your manuscript is conditional on all authors' agreement with our publication policies (see <https://www.nature.com/natcomputsci/for-authors>). In particular

your manuscript must not be published elsewhere and there must be no announcement of the work to any media outlet until the publication date (the day on which it is uploaded onto our web site).

Before your manuscript is typeset, we will edit the text to ensure it is intelligible to our wide readership and conforms to house style. We look particularly carefully at the titles of all papers to ensure that they are relatively brief and understandable.

Once your manuscript is typeset, you will receive a link to your electronic proof via email with a request to make any corrections within 48 hours. If, when you receive your proof, you cannot meet this deadline, please inform us at rjsproduction@springernature.com immediately.

If you have queries at any point during the production process then please contact the production team at rjsproduction@springernature.com. Once your paper has been scheduled for online publication, the Nature press office will be in touch to confirm the details.

Content is published online weekly on Mondays and Thursdays, and the embargo is set at 16:00 London time (GMT)/11:00 am US Eastern time (EST) on the day of publication. If you need to know the exact publication date or when the news embargo will be lifted, please contact our press office after you have submitted your proof corrections. Now is the time to inform your Public Relations or Press Office about your paper, as they might be interested in promoting its publication. This will allow them time to prepare an accurate and satisfactory press release. Include your manuscript tracking number NATCOMPUTSCI-23-0044B and the name of the journal, which they will need when they contact our office.

About one week before your paper is published online, we shall be distributing a press release to news organizations worldwide, which may include details of your work. We are happy for your institution or funding agency to prepare its own press release, but it must mention the embargo date and Nature Computational Science. Our Press Office will contact you closer to the time of publication, but if you or your Press Office have any inquiries in the meantime, please contact press@nature.com.

We welcome the submission of potential cover material (including a short caption of around 40 words) related to your manuscript; suggestions should be sent to Nature Computational Science as electronic files (the image should be 300 dpi at 210 x 297 mm in either TIFF or JPEG format). We also welcome suggestions for the Hero Image, which appears at the top of our <http://www.nature.com/natcomputsci> home page; these should be 72 dpi at 1400 x 400 pixels in JPEG format. Please note that such pictures should be selected more for their aesthetic appeal than for their scientific content, and that colour images work better than black and white or grayscale images. Please do not try to design a cover with the Nature Computational Science logo etc., and please do not submit composites of images related to your work. I am sure you will understand that we cannot make any promise as to whether any of your suggestions

might be selected for the cover of the journal.

Best,
Fernando

--

Fernando Chirigati, PhD
Chief Editor, Nature Computational Science
Nature Portfolio

P.S. Click on the following link if you would like to recommend Nature Computational Science to your librarian: https://www.springernature.com/gp/librarians/recommend-to-your-library

** Visit the Springer Nature Editorial and Publishing website at www.springernature.com/editorial-and-publishing-jobs for more information about our career opportunities. If you have any questions please click here.**